

# Assessing the large-scale impacts of environmental change using a coupled hydrology and soil erosion model

Joris P.C. Eekhout[1], Wilco Terink[2], and Joris de Vente[1,3]

[1]Soil Erosion and Conservation Research Group, CEBAS-CSIC, Spanish Research Council, Campus Universitario Espinardo, 30100, P.O. Box 164, Murcia, Spain
[2]IQ-Hydrology, Ben van Londenstraat 48, 6709 TM Wageningen, The Netherlands
[3]FutureWater, Costerweg 1V, 6702 AA, Wageningen, The Netherlands

**Correspondence:** Joris Eekhout (joriseekhout@gmail.com)

**Abstract.** Assessing the impacts of environmental change on soil erosion and sediment yield at the large catchment scale remains one of the main challenges in soil erosion modelling studies. Here, we present a process-based soil erosion model, based on the integration of the Morgan-Morgan-Finney erosion model in a daily-based hydrological model. The model overcomes many of the limitations of previous large-scale soil erosion models, as it includes a more complete representation of crucial

processes like surface runoff generation, dynamic vegetation development, and sediment deposition, and runs at the catchment scale with a daily time step. This makes the model especially suited for evaluation of the impacts of environmental change on soil erosion and sediment yield at large spatial scales. The model was successfully applied in a large catchment in southeastern Spain. We demonstrate the models capacity to perform impact assessments of environmental change scenarios, specifically simulating the scenario impacts of intra- and inter-annual variations in climate, land management and vegetation development

on soil erosion and sediment yield.

*Copyright statement.* TEXT

## 1 Introduction

Climate change will likely affect soil erosion and sediment yield across scales (e.g. Nearing et al., 2004; Li et al., 2006; Burt et al., 2016). However, assessing the impacts of environmental change on soil erosion and sediment yield at the large catchment

scale remains one of the main challenges in soil erosion modelling studies (de Vente et al., 2013; Poesen, 2018). Most soil erosion and sediment yield models adopt simplified model formulations, are applied at low spatial and temporal resolutions, and often only partly represent the impacts of changes in land use or climate conditions. This often leads to unreliable results that do not sufficiently increase process understanding or support decision-making. To overcome part of these limitations, here, we present a process-based, large-scale, soil erosion model, coupled to a hydrological model, accounting for the most

relevant factors determining soil erosion by water, including saturated and infiltration excess surface runoff, dynamic vegetation development and sediment deposition.



First of all, soil erosion by water occurs by the impact of raindrops and by the flow of water on the soil surface or the river bed. It is therefore crucial to quantify raindrop impact, overland flow and possible interactions with vegetation cover. Soil erosion by the impact of raindrops is a function of the amount and the size of the raindrops that reach the soil surface. Vegetation cover can reduce the impact of raindrops by interception, separating the precipitation into direct throughfall, with a high impact, and

leaf drainage, with a lower impact. Assessment of soil erosion, therefore, needs to account for spatial and temporal changes in vegetation cover. Nevertheless, most large-scale soil erosion assessments do not consider dynamic seasonal and inter-annual vegetation development, while previous studies have shown that cropping patterns, inter-annual trends and land use changes can have a significant impact on soil erosion rates (e.g. Maetens et al., 2012).

Soil erosion is also a function of runoff. Runoff generation depends on surface and sub-surface processes and is a function of

precipitation volume and intensity, soil moisture and soil hydraulic properties. Process-based models often incorporate a separate hydrological model to simulate surface runoff generation, which then directly forces soil erosion by runoff. Surface runoff may be generated by several distinct processes, from which saturation excess and infiltration excess are the most common processes Beven (2012). Saturation excess surface runoff occurs when the soil water content reaches saturation, while infiltration excess surface runoff occurs when the precipitation intensity exceeds the soils infiltration capacity. However, many large-scale

soil erosion models only consider saturation excess surface runoff, disregarding the infiltration excess surface runoff mechanism. Infiltration excess surface runoff is a sub-daily process that is often only implemented in event-scale models. Infiltration excess surface runoff is an especially important process in areas where a major part of soil erosion takes place during extreme rainfall events with high precipitation intensities (Mulligan, 1998; López-Bermúdez et al., 2002; Farnsworth and Milliman, 2003; González-Hidalgo et al., 2007; Gonzalez-Hidalgo et al., 2012).

Furthermore, assessing catchment sediment yield requires evaluation of the sediment transport capacity and sediment deposition. Sediment deposition occurs when the sediment transport capacity of the runoff is exceeded and basically depends on the interaction between the texture of the detached soil material, flow velocity and the roughness of the surface. Large particles are more likely to be deposited close to the source, while small particles are more easily brought and maintained into transport. The roughness of the surface is a combination of the roughness of the soil surface and the roughness caused by vegetation.

Many studies have highlighted the importance of accounting for sediment transport and sediment deposition, claiming that often a large part of the eroded sediment is deposited close to its source (Walling, 1983; de Vente et al., 2007). However, many large-scale soil erosion models insufficiently consider this process or even only simulate soil detachment processes.

From the available soil erosion models, process-based models aim to incorporate the most relevant processes driving soil detachment, sediment transport and deposition, as described in the previous paragraphs, and often run at small spatial (hillslope

to small catchment) and temporal scales (hourly to daily time steps). Most detailed assessments are obtained from event-scale model applications in process-based models, such as WEPP (Nearing et al., 1989), which require detailed observational input data, such as (sub-)hourly precipitation and detailed spatial information, and incorporate a large number of model calibration parameters. While these models present a strong potential to provide increased process understanding, it is often unfeasible to obtain all required input data for large catchments. Furthermore, the high uncertainty on how input data and model parameters

will change under scenarios of environmental change severely limits their application in large-scale assessments.



At these scales, soil erosion is often assessed using so-called empirical erosion models. These models are derived from field studies where soil erosion has been observed under different land use, management, soil, climate, and topographical conditions. The most well-known and applied empirical model is the Universal Soil Loss Equation (USLE) Wischmeier and Smith (1978) and its derivatives RULSE Renard et al. (1997) and MUSLE Williams (1995). While the empirical formulations of the USLE

were obtained at plot-scale, the model is often applied at much larger scales, sometimes in combination with a sediment transport capacity equation or a sediment delivery ratio to assess sediment yield. Due to its simplicity, the USLE can be applied with a relatively limited amount of input data. However, their main restriction is the limited number of processes accounted for (e.g. the USLE and RUSLE based models only consider sheet and rill erosion) and the limited potential to evaluate the impacts of changes in climate and land management. Furthermore, these models are typically applied at annual time steps,

largely neglecting intra-annual variation of climate and vegetation conditions.

Most current soil erosion models have a limited potential for application at larger temporal and spatial scales (i.e. process-based models) or lack sufficient representation of the underlying soil detachment and sediment transport processes and sensitivity to changes in land use or climate (i.e. empirical models), making them of limited use for scenario studies and process understanding. Here, we present a process-based soil erosion model based on the integration of the Morgan-Morgan-Finney

erosion model (MMF; Morgan and Duzant, 2008) and the spatially distributed hydrological model Spatial Processes in HYdrology (SPHY; Terink et al., 2015). This integrated model overcomes many of the limitations of previous large-scale soil erosion models, as it includes a more complete representation of crucial processes like surface runoff generation, dynamic vegetation development, and sediment deposition, and runs at the catchment scale with a daily time step. This makes the model especially suitable for evaluation of the inter- and intra-annual impacts of environmental change on soil erosion and sediment yield at

large spatial scales. In the next paragraphs we first present the different model components and enhancements as compared to previous models. Then we illustrate its functionality and potential for scenario studies by application to the Upper Segura catchment in southeastern Spain under present and projected future climate conditions.

## 2   Model Description

### 2.1   Model Overview

The SPHY-MMF model presented here is an integration of the (Modified) Morgan-Morgan-Finney soil erosion model into the SPHY hydrological model (version 2.1). Figure 1 shows the main hydrological and soil erosion processes considered by the model. SPHY is a spatially distributed leaky-bucket type model that simulates hydrological processes on a cell-by-cell basis at a daily timestep (Terink et al., 2015). The model is written in the Python programming language using the PCRaster dynamic modelling framework (Karssenberg et al., 2010). MMF is a conceptual soil erosion model that originally is applied with an

annual time step. Here we present a modification of the model at a daily time step, fully integrated with the SPHY model. MMF receives input from the SPHY model, such as effective precipitation (throughfall), runoff and canopy cover for calculation of erosion and deposition processes.





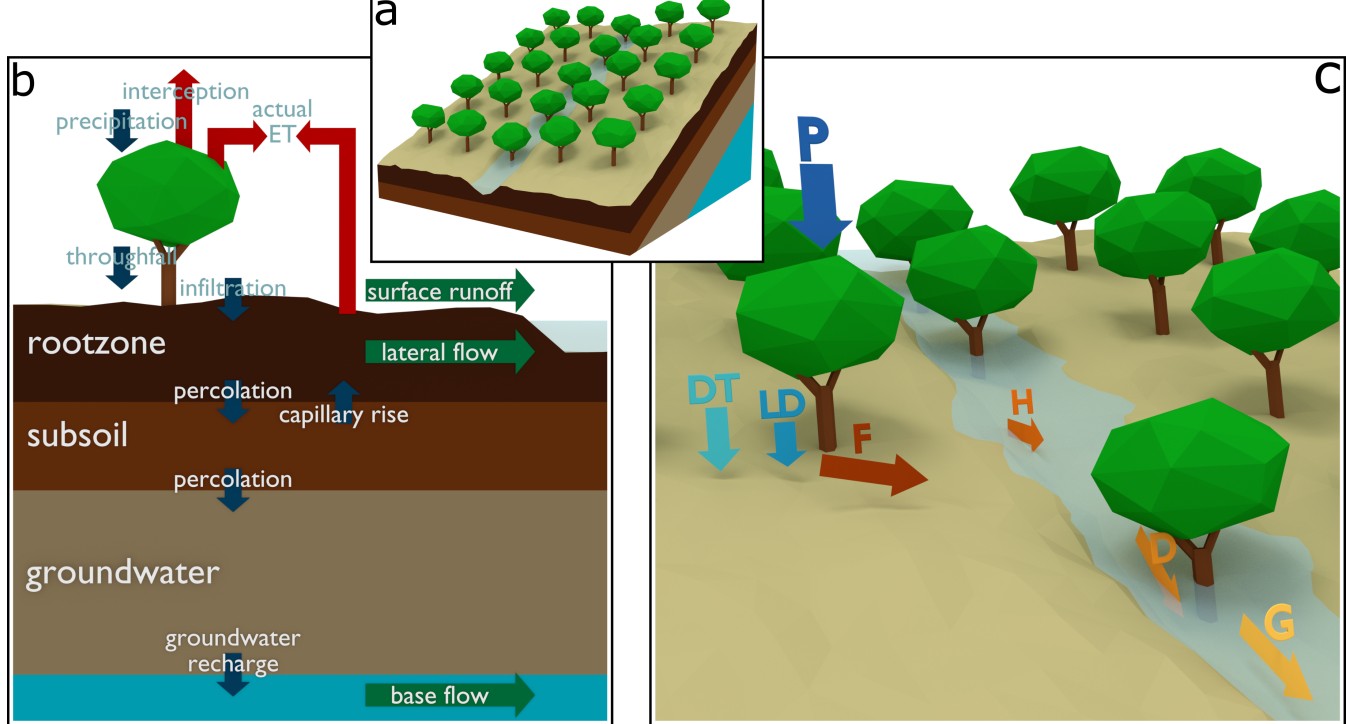

**Figure 1.** Overview of the model: (a) representation of a single cell, (b) the hydrological processes, and (c) the soil erosion processes.

## 2.2 Hydrological model

SPHY simulates most relevant hydrological processes (Figure 1b), such as interception, evapotranspiration, dynamic evolution of vegetation cover, surface runoff, and lateral and vertical soil moisture flow at a daily time step. The model is described in full detail by Terink et al. (2015), therefore, here we only provide a summary of the processes that are simulated by the model,

some hydrological processes that have been changed with respect to the original SPHY model, and a detailed description of the processes that are related to the integration of MMF.

SPHY requires daily precipitation and temperature maps as input. Effective precipitation is determined by subtracting canopy storage and interception from precipitation. Canopy storage is determined from the Leaf Area Index (LAI), which is derived from Normalized Differenced Vegetation Index (NDVI) images. Reference evapotranspiration is determined using the Harg-

reaves equation (Hargreaves and Samani, 1985), which is subsequently multiplied by the crop coefficient to obtain the potential evapotranspiration. The crop coefficient is determined from the NDVI images using a linear relationship. Actual evapotranspiration is obtained by multiplying the potential evapotranspiration by a reduction factor for water deficit or water surplus, which are functions of current soil water content, soil hydraulic properties and plant-specific water need. Surface runoff is determined by a daily implementation of the Green-Ampt formula and is a function of infiltration, effective precipitation and soil hydraulic

properties. The soil profile consists of three layers, i.e. rootzone, subzone and groundwater layer. Water can percolate from





the rootzone to the subzone and from the subzone to the groundwater layer. Water travels from the subzone to the rootzone through capillary rise. Water drains from the rootzone as lateral flow and from the groundwater layer as baseflow. The total runoff is the sum of surface runoff, lateral flow and baseflow. All soil processes are functions of current water content (in the respective layers) and soil hydraulic properties, i.e. saturated hydraulic conductivity, saturated water content, field capacity and

wilting point. Water is routed using a single flow algorithm. A flow recession coefficient accounts for flow delay from channel friction. When reservoirs are present, the user can opt to include an advanced routing scheme accounting for reservoir storage and outflow.

### 2.2.1   Evapotranspiration

The actual evapotranspiration $ET_a$ is determined by multiplying the potential evapotranspiration with the reduction parameters

for water surplus and water deficit conditions. In the current version of the hydrological model we have changed the reduction parameter for water shortage conditions by the method proposed by Allen et al. (1998):

$$K_s = \frac{TAW - D_r}{(1 - p)TAW} \tag{1}$$

Where $TAW$ is the total available water in the rootzone (mm), $D_r$ the root zone depletion (mm) and $p$ the depletion fraction (-). The total available water $TAW$ is defined as:

$$TAW = \theta_{\text{FC}} - \theta_{\text{WP}} \tag{2}$$

Where $\theta_{\text{FC}}$ is the soil water content at field capacity (mm) and $\theta_{\text{WP}}$ the soil water content at wilting point (mm). The root zone depletion $D_r$ is defined as:

$$D_r = \theta_{\text{FC}} - \theta \tag{3}$$

Where $\theta$ is the current soil water content (mm). The depletion fraction $p$ is defined as the fraction of $TAW$ that a crop can

extract from the root zone without suffering water stress, which is determined by the following equation:

$$p = p_{\text{tabular}} + 0.04(5 - ET_{\text{pot}}) \tag{4}$$

Where $p_{\text{tabular}}$ is a landuse-specific tabular value of the depletion fraction (-) and $ET_{\text{pot}}$ is the potential evapotranspiration (mm). Values for the landuse-specific tabular value of the depletion fraction can be obtained from Allen et al. (1998) (Table 22).

Open-water evaporation is determined in the reservoir cells. In these cells all soil hydraulic processes are turned off and

runoff equals precipitation. Open-water evaporation is determined as follows and assumes presence of water in the reservoir cells:

$$ET_{\text{open-water}} = kc_{\text{open-water}}ET_{\text{ref}} \tag{5}$$

Where $kc_{\text{open-water}}$ is the crop coefficient value for open-water evaporation (-) and $ET_{\text{ref}}$ is the reference evapotranspiration (mm). We set $kc_{\text{open-water}}$ to a value of 1.2, after Allen et al. (1998). In each time step the open-water evaporation is subtracted

from the reservoir storage.



### 2.2.2 Infiltration excess surface runoff

The original SPHY model simulates saturated surface runoff but not infiltration excess surface runoff. Therefore, we have incorporated an infiltration excess equation at a daily time step based on the Green-Ampt formula Heber Green and Ampt (1911). We assumed a constant infiltration rate $f$ ($\mathrm{mm\,hr^{-1}}$), which is determined for each cell and each day by:

$$f = \frac{K_{\mathrm{eff}}}{24}\left[1 + \frac{\theta_{\mathrm{sat}} - \theta}{\theta_{\mathrm{sat}}}\right]^{\lambda} \tag{6}$$

Where $K_{\mathrm{eff}}$ is the effective hydraulic conductivity ($\mathrm{mm\,day^{-1}}$), $\theta_{\mathrm{sat}}$ is the saturated water content ($\mathrm{mm}$), $\theta$ is the actual water content ($\mathrm{mm}$), and $\lambda$ is a calibration parameter (-). Bouwer (1969) suggested an approximation of $K_{\mathrm{eff}} \approx 0.5 K_{\mathrm{sat}}$. We included a calibration parameter $k$ to be able to change the value of $K_{\mathrm{eff}}$ as a fraction of $K_{\mathrm{sat}}$ ($K_{\mathrm{eff}} = k K_{\mathrm{sat}}$).

Infiltration excess surface runoff occurs when the precipitation intensity exceeds the infiltration rate $f$ Beven (2012). Analysis of hourly precipitation time series for 25 years (1991-2015) from 5 precipitation stations in a large catchment in southeastern Spain showed that, on average, the highest precipitation intensity was recorded in the first hour of the rain storm and decreases linearly until the end of the storm. We assumed a triangular-shaped precipitation intensity $p(t)$ ($\mathrm{mm\,hr^{-1}}$) according to:

$$p(t) = -\frac{1}{2}\alpha^2 P t + \alpha P \tag{7}$$

Where $\alpha$ is the fraction of daily rainfall that occurs in the hour with the highest intensity (-), $P$ is the daily rainfall ($\mathrm{mm}$), and $t$ is an hourly time step. Daily infiltration excess surface runoff $Q_{\mathrm{surf}}$ is determined as:

$$Q_{\mathrm{surf}} = \begin{cases} \dfrac{(\alpha P - f)^2}{\alpha^2 P} & \text{if } \alpha P > f \\ 0 & \text{if } \alpha P \leq f \end{cases} \tag{8}$$

### 2.2.3 Dynamic vegetation processes

SPHY-MMF contains a dynamic vegetation module that allows characterization of the seasonal and inter-annual differences in vegetation cover and resulting canopy storage, interception and precipitation throughfall. The latter is subsequently used in both the hydrological and soil erosion model. A time series of the NDVI images is used as input for the dynamic vegetation module. NDVI images may only be available for a limited period, e.g. from 2000 - present for MODIS NDVI images (Moderate Resolution Imaging Spectroradiometer; Didan, 2015). Therefore, in the Model Application section we present a method to obtain NDVI images for historical and future model assessments. The Leaf Area Index (LAI) is determined from the individual NDVI images using the following logarithmic relation, which is valid for vegetation that is evenly distributed over a surface (Sellers et al., 1996):

$$LAI = LAI_{\mathrm{max}} \frac{\log(1 - FPAR)}{\log(1 - FPAR_{\mathrm{max}})} \tag{9}$$

Where $LAI_{\mathrm{max}}$ is the maximum LAI (-), $FPAR$ is the photosynthetically active radiation (-) and $FPAR_{\mathrm{max}}$ is the maximum $FPAR$ (-), which is set to 0.95 (Sellers et al., 1996). The maximum LAI $LAI_{\mathrm{max}}$ is vegetation dependent, values for several



vegetation types can be found in Sellers et al. (1996). The photosynthetically active radiation $FPAR$ is determined as follows:

$$FPAR = \frac{(SR - SR_{\min})(FPAR_{\max} - FPAR_{\min})}{SR_{\max} - SR_{\min}} + FPAR_{\min} \tag{10}$$

Where $SR$ is a transformation of NDVI (-), $SR_{\min}$ and $SR_{\max}$ are the minimum and maximum $SR$ values (-), respectively, and $FPAR_{\min}$ is the minimum $FPAR$ (-), which is set to 0.001 (Sellers et al., 1996). $FPAR$ is bounded by $FPAR_{\min}$ and $FPAR_{\max}$. $SR$ is determined as follows:

$$SR = \frac{1 + NDVI}{1 - NDVI} \tag{11}$$

$SR_{\min}$ and $SR_{\max}$ are determined with Equation 11, applying an NDVI value corresponding to the 5% and 98% quantiles, respectively.

## 2.3 Soil erosion simulation with a daily-based Morgan-Morgan-Finney model

The soil erosion model is based on the Modified MMF model (Morgan and Duzant, 2008). In the current model (Figure 1c), total soil erosion is calculated from detachment by raindrop impact and detachment by runoff, while sediment yield is calculated by routing detached sediment, considering within cell deposition and sediment transport capacity. Detachment of soil particles from raindrop impact is determined from the total rainfall energy, which is determined for direct throughfall and leaf drainage, respectively. Detachment of soil particles by runoff is determined from the accumulated runoff from the hydrological model. Both soil erosion equations account for the fraction of the soil covered by stones and vegetation or snow and are determined separately for three texture classes (sand, silt, clay). Within cell deposition is calculated as a function of vegetation and surface roughness. The remainder of the detached sediment is taken into transport and routed through the catchment, taking into account the transport capacity of the flow and the trapping efficiency of the reservoirs. The next paragraphs provide a detailed description of all these processes.

### 2.3.1 Estimation of rainfall energy

The total kinetic energy of the effective precipitation ($KE$, $\mathrm{J\,m^{-2}}$) is used to determine the detachment of soil particles by raindrop impact and is defined as:

$$KE = KE_{DT} + KE_{LD} \tag{12}$$

Where $KE_{LD}$ is the kinetic energy of the leaf drainage ($\mathrm{J\,m^{-2}}$) and $KE_{DT}$ is the kinetic energy of the direct throughfall ($\mathrm{J\,m^{-2}}$).

The kinetic energy of the leaf drainage is based on Brandt (1990):

$$KE_{LD} = \begin{cases} 0 & \text{for } PH < 0.15 \\ LD(15.8PH^{0.5} - 5.87) & \text{for } PH \geq 0.15 \end{cases} \tag{13}$$



Where $LD$ is the leaf drainage (mm) and $PH$ is the plant height (m), specified for each landuse class.

The kinetic energy of the direct throughfall is based on a relationship described by Marshall and Palmer (1948), which is representative of a wide range of environments (Morgan, 2005):

$$KE_{DT} = DT(8.95 + 8.44 \log_{10} I) \tag{14}$$

Where $DT$ is the direct throughfall (mm) and $I$ is the intensity of the erosive precipitation ($\mathrm{mm\,h^{-1}}$). The intensity of the erosive precipitation is a model parameter and varies according to geographical location. Morgan and Duzant (2008) proposes $10\ \mathrm{mm\,h^{-1}}$ for temperate climates, $25\ \mathrm{mm\,h^{-1}}$ for tropical climates and $30\ \mathrm{mm\,h^{-1}}$ for strongly seasonal climates (e.g. Mediterranean, tropical monsoon).

The leaf drainage $LD$, i.e. precipitation that reaches the soil surface as flow or drips from the leaves and stems of the vege-
tation, and direct throughfall $DT$, i.e. precipitation that reaches the soil surface directly through gaps in the vegetation cover, from equations 13 and 14, are obtained from the effective precipitation ($P_{\mathrm{eff}}$, mm). The effective precipitation (throughfall, $P_{\mathrm{eff}}$, mm) from the hydrological model is first corrected for the slope angle, following Choi et al. (2017):

$$P_{\mathrm{eff}} = P_{\mathrm{eff}} \cos S \tag{15}$$

Where $P_{\mathrm{eff}}$ is the effective precipitation (mm) and $S$ the slope (°).

Leaf drainage is determined as:

$$LD = P_{\mathrm{eff}} CC \tag{16}$$

Where $CC$ is the canopy cover (proportion between zero and unity). The canopy cover is either introduced by a landuse-class specific tabular value or determined by the vegetation module. When the vegetation module is used, the canopy cover is obtained from the LAI (Equation 9), maximized by 1.

Direct throughfall becomes:

$$DT = P_{\mathrm{eff}} - LD \tag{17}$$

### 2.3.2   Detachment of soil particles

Detachment of soil particles is determined separately for raindrop impact and accumulated runoff and is subsequently summed.

The detachment of soil particles by raindrop impact ($F$, $\mathrm{kg\,m^{-2}}$) and the detachment of soil particles by runoff ($H$, $\mathrm{kg\,m^{-2}}$)
are determined for each of the soil texture classes separately and subsequently summed. The detachment of soil particles by raindrop impact is calculated as:

$$F_i = K_i \frac{\%i}{100}(1 - GC) KE \times 10^{-3} \tag{18}$$

With $K$ the detachability of the soil ($\mathrm{g\,J^{-1}}$), $i$ the textural class, with $c$ for clay, $z$ for silt and $s$ for sand, and $GC$ the ground cover (-). The detachability of the soil for each texture class is included as a model parameter, for which Quansah (1982)





proposed $K_c = 0.1$, $K_z = 0.5$ and $K_s = 0.3$ g J$^{-1}$. The ground cover, expressed as a proportion between zero and unity, protects the soil from detachment and is determined by the proportion of vegetation and rocks covering the surface. The ground cover is set to 1 in case the surface is covered with snow, which is determined in the hydrological model.

The detachment of soil particles by runoff ($H$, kg m$^{-2}$) is calculated as:

$$H_i = DR_i \frac{\%i}{100} Q^{1.5}(1 - GC)\sin^{0.3} S \times 10^{-3} \tag{19}$$

Where $Q$ is the volume of accumulated runoff (mm) and $DR$ the detachability of the soil by runoff (g mm$^{-1}$). The detachability of the soil for each texture class is included as a model parameter for which Quansah (1982) proposed $DR_c = 1.0$, $DR_z = 1.6$ and $DR_s = 1.5$ g mm$^{-1}$.

### 2.3.3 Immediate deposition of detached particles

A proportion of the detached soil is deposited in the cell of its origin and is a function of the abundance of vegetation and the surface roughness. The percentage of the detached sediment that is deposited ($DEP$) is estimated from the relationship obtained by Tollner et al. (1976) and calculated separately for each texture class:

$$DEP_i = 44.1 N_{f_i}^{0.29} \tag{20}$$

Where $N_f$ is the particle fall number (-), defined as:

$$N_{f_i} = \frac{l v_{s_i}}{v d} \tag{21}$$

Where $l$ is the length of a grid cell (m), $v_s$ the particle fall velocity (m s$^{-1}$), $v$ the flow velocity (m s$^{-1}$) and $d$ the depth of flow (m). Particle fall velocities are estimated from:

$$v_s = \frac{1/18\delta^2(\rho_s - \rho)g}{\eta} \tag{22}$$

Where $\delta$ is the diameter of the particle (m), $\rho_s$ the sediment density (= 2650 kg m$^{-3}$), $\rho$ the flow density (typically 1100 kg m$^{-3}$ for runoff on hillslopes; Abrahams et al., 2001), $g$ gravitational acceleration (taken as 9.81 m s$^{-2}$) and $\eta$ the fluid viscosity (nominally 0.001 kg m$^{-1}$ s$^{-1}$ but taken as 0.0015 to allow for the effects of the sediment in the flow; Morgan and Duzant, 2008). When Equation 22 is applied to the three texture sizes of 2 μm for clay, 60 μm for silt and 200 μm for sand, this gives respective values of 2 $10^{-6}$ m s$^{-1}$ for clay, 2 $10^{-3}$ m s$^{-1}$ for silt and 0.02 m s$^{-1}$ for sand.

The flow velocity $v$ from Equation 21 is obtained by the Manning formula:

$$v = \frac{1}{n'} d^{2/3} S^{1/2} \tag{23}$$

Where $n'$ is the modified Manning's roughness coefficient (s m$^{-1/3}$), which is a combination of the Manning's roughness coefficient for the soil surface and vegetation, defined as (Petryk and Bosmajian, 1975):

$$n' = \sqrt{n_{\text{soil}}^2 + n_{\text{vegetation}}^2} \tag{24}$$





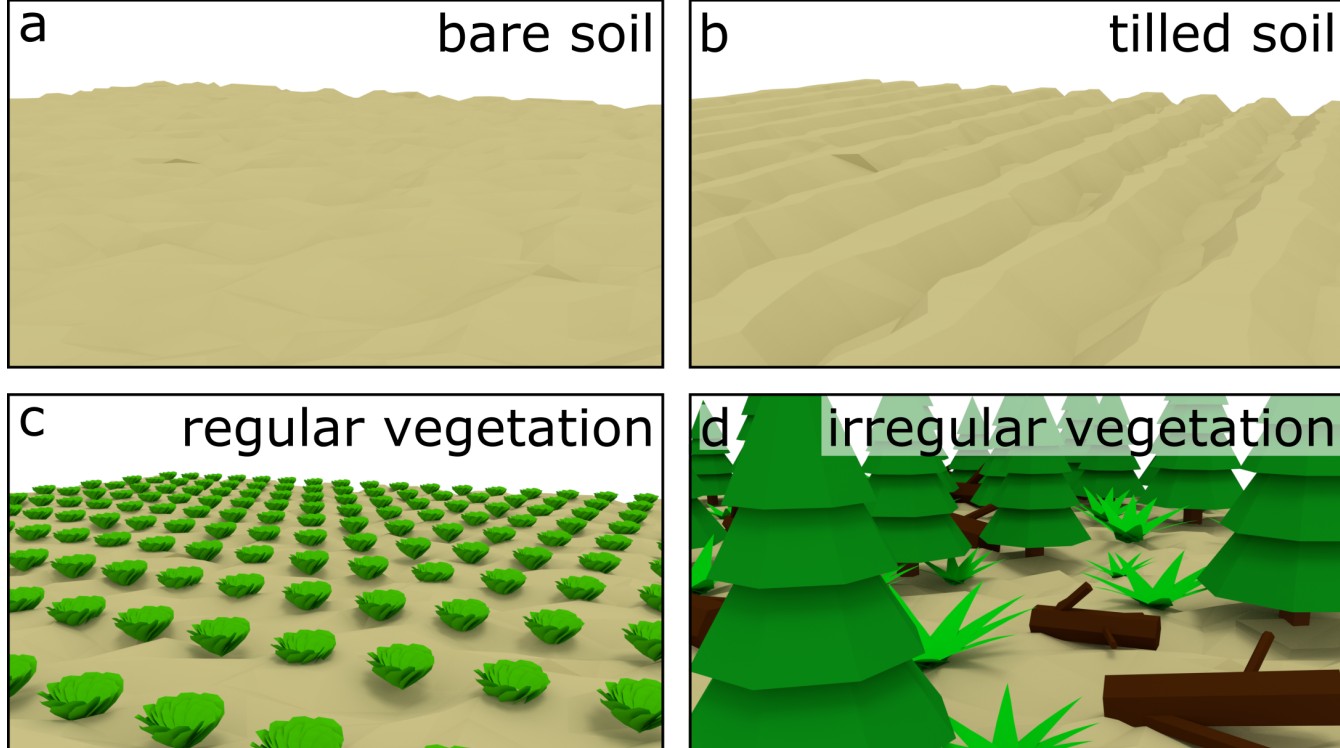

**Figure 2.** Surface and vegetation roughness options: (a) bare soil, (b) tilled soil, (c) regular vegetation, and (d) irregular vegetation.

The Manning's roughness coefficient for bare soil $n_{\text{soil}}$ is set to $0.015\,\text{s}\,\text{m}^{-1/3}$, as suggested by Morgan and Duzant (2008) (Figure 2a). For tilled conditions (Figure 2b) the following equation is applied:

$$n_{\text{soil}} = \exp(-2.1132 + 0.0349 RFR) \tag{25}$$

Where $RFR$ is the surface roughness parameter ($\text{cm}\,\text{m}^{-1}$). Values for $RFR$ are tillage implementation specific and can be
5  obtained from Morgan and Duzant (2008) (Table IV).

The Manning's roughness coefficient for regular spaced vegetation (Figure 2c) $n_{\text{vegetation}}$ is obtained from the following equation (Jin et al., 2000):

$$n_{\text{vegetation}} = \frac{d^{2/3}}{\sqrt{\frac{2g}{DNV}}} \tag{26}$$

Where $D$ is the stem diameter (m) and $NV$ the stem density ($\text{stems}\,\text{m}^{-2}$). Stem diameter and stem density may be difficult
10  to obtain for certain landuse classes with irregular spaced vegetation (e.g. forest, shrubland), therefore, users may opt to use tabular values for $n_{\text{vegetation}}$, e.g. from Chow (1959) (Figure 2d). Equation 26 results in unrealistically high flow velocity values for landuse classes where the stem density is very low, such as in orchards. Therefore, in these conditions where the influence of vegetation on flow velocity is negligible, $n_{\text{vegetation}}$ can be set to 0.



Equations 21, 23 and 26 require a flow depth $d$, a model parameter that can be used in the model calibration. The value for $d$ should be taken such that it corresponds to a water depth from runoff generated within the cell margins, i.e. without accumulation of flow from upstream located cells.

### 2.3.4 Sediment deposition and transport

The amount of sediment that is routed to downstream cells is determined from the sum of the detached sediment from raindrop impact (Equation 18) and accumulated runoff (Equation 19), subtracting the proportion of the sediment that is deposited within the cell of its origin (Equation 20):

$$G = (F_i + H_i)(1 - (DEP_i/100)) \tag{27}$$

The amount of sediment that is routed to downstream cells is the summation of the individual amounts for clay, silt and sand.

Sediment is routed using a routing scheme that takes into account both the transport capacity ($TC$; $\mathrm{ton\,ha^{-1}}$) of the accumulated runoff and the trapping efficiency of the reservoirs ($TE$; -). The transport capacity $TC$ (Figure 3a) of the accumulated runoff is based on Prosser and Rustomji (2000):

$$TC = \mathrm{flow_{factor}} q_{surf}^{\beta} S^{\gamma} \tag{28}$$

Where $\mathrm{flow_{factor}}$ is a spatially distributed roughness factor (-), $q_{surf}$ accumulated runoff per unit width ($\mathrm{m^2\,day^{-1}}$), $S$ the local

energy gradient (°), approximated by the slope, and $\beta$ and $\gamma$ are model parameters (-). As suggested by Prosser and Rustomji (2000) $\gamma = 1.4$ and $\beta$ is used for model calibration.

The roughness factor $\mathrm{flow_{factor}}$ is determined as follows:

$$\mathrm{flow_{factor}} = \frac{v_{\mathrm{actual}}}{v_{\mathrm{b}}} \tag{29}$$

Where $v_{\mathrm{actual}}$ is the actual flow velocity ($\mathrm{m\,s^{-1}}$) and $v_{\mathrm{b}}$ is the flow velocity for bare soil conditions ($\mathrm{m\,s^{-1}}$). The actual flow

velocity $v_{\mathrm{actual}}$ is obtained from Equations 23-26, applying a water depth $d$ of 0.25 m, which coincides with deeper rills from Morgan and Duzant (2008). The flow velocity for bare soil conditions $v_{\mathrm{b}}$ is obtained from Equation 23, applying values for $n' = 0.015\,\mathrm{s\,m^{-1/3}}$ and $d = 0.005$ m (Morgan and Duzant, 2008).

Reservoir sediment trapping efficiency $TE$ (Figure 3b), the percentage of sediment trapped by the reservoir, is calculated according to Brown (1943):

$$TE = 100 \left[ 1 - \frac{1}{1 + 0.0021 D \frac{C}{A_{\mathrm{basin}}}} \right] \tag{30}$$

Where $D$ is a constant (-) within the range 0.046-1, depending on the reservoir operation characteristics that we set at 0.1, $C$ the reservoir capacity ($\mathrm{m^3}$), and $A_{\mathrm{basin}}$ the drainage area of the subcatchment ($\mathrm{km^2}$).





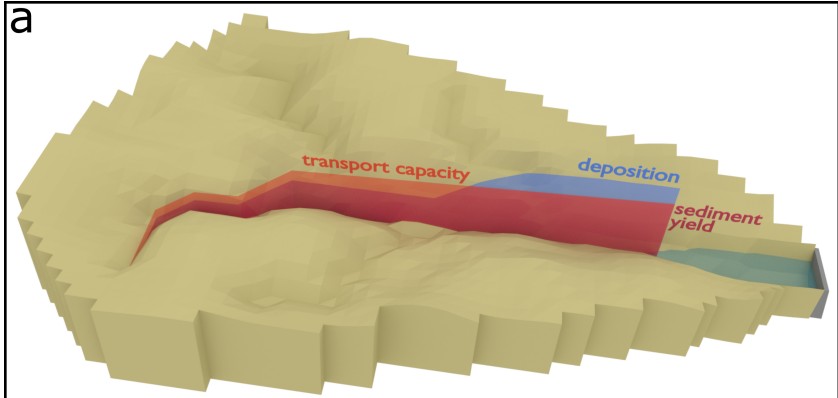

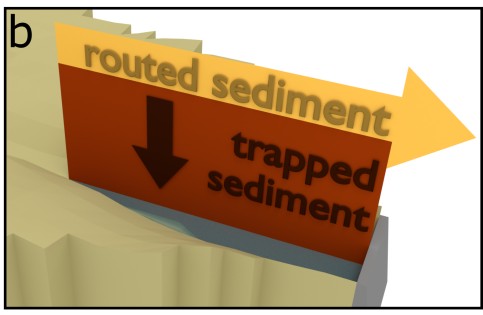

**Figure 3.** Sediment routing: (a) transport of sediment through the catchment, and (b) trapping efficiency at the reservoirs.

## 3   Model Application

To illustrate the model performance, its functionality and capacity for use in scenario studies, we applied the model to the Upper Segura River catchment (2,589 $km^2$) under present and future projected climate conditions. The Upper Segura catchment is located in the headwaters of the Segura River in southeastern Spain (Figure 4). The elevation ranges between 411 and 2055

m.a.s.l. (Figure 4). The climate in the catchment is classified as temperate (Cfa and Cfb according to the Köppen-Geiger climate classification, 80%) and semi-arid (BSk, 20%). The catchment-average annual precipitation is 570 mm (for the period 1981-2000) and the mean annual temperature is 13.2 °C (1981-2000).

The main landuse types are forest (45%), shrubland (40%), cereal fields (7%) and almond orchards (4%) (Figure 4), based on a detailed landuse map Ministerio de Agricultura y Pesca Alimentación y Medio Ambiente (2010). Agriculture accounts

for 14% of the catchments surface area. Huerta is defined as small-scale traditional vegetable and/or fruit orchards, which are common in the study area. The main soil classes are Leptosols (38%), Luvisols (27%), Cambisols (16%) and Calcisols (11%), based on the SoilGrids database Hengl et al. (2017). There are 5 reservoirs located in the catchment (Figure 4b) with a total capacity of 663 $Hm^3$, which are mainly used to store water for irrigation purposes.

### 3.1   Input Data

All input data were prepared at a 200 m grid size. Daily precipitation data were obtained from the SPREAD dataset Serrano-Notivoli et al. (2017), with a 5 km spatial resolution. Daily temperature data were obtained from the SPAIN02 dataset Herrera et al. (2016), with a 0.11°resolution. Precipitation and temperature data were subsequently interpolated on the model grid using bivariate interpolation (Akima, 1996). Soil textural fractions (sand, clay and silt) and soil organic matter content were obtained from the global SoilGrids dataset (Hengl et al., 2017) at 250 m resolution. The soil hydraulic properties (saturated

hydraulic conductivity, saturated water content, field capacity, and wilting point) were obtained by applying pedotransfer functions (Saxton and Rawls, 2006). A Digital Elevation Model was obtained from the SRTM dataset (Farr et al., 2007) at 30



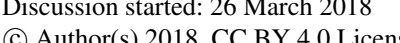

**Figure 4.** Location and characteristics of the Upper Segura River catchment: (a) location of the catchment within Europe, (b) the hydrological calibration area (orange), the channels (light blue), the reservoirs (dark blue), and the calibration reservoirs (red dots), (c) Digital Elevation Model (Farr et al., 2007), (d) landuse map (Ministerio de Agricultura y Pesca Alimentación y Medio Ambiente, 2010), and (e) soil texture map (Hengl et al., 2017).

m resolution and was resampled to the model grid (Figure 4d). The spatially distributed rock fraction map was obtained by applying the empirical formulations from Poesen et al. (1998), which determines rock fraction based on slope gradient.

Both the hydrological and the soil erosion model require landuse-specific input. We used a detailed landuse map (Ministerio de Agricultura y Pesca Alimentación y Medio Ambiente, 2010) that identifies 25 landuse classes within the study area. Values

5    for the landuse-specific tabular value of the depletion fraction $p_{\mathrm{tabular}}$ to calculate actual evapotranspiration were obtained from Allen et al. (1998) (Table 22). Values for the maximum LAI ($LAI_{\mathrm{max}}$) were obtained from Sellers et al. (1996). The soil erosion model requires landuse-specific input for plant height ($PH$), stem density ($NV$), stem diameter ($D$), ground





**Table 1.** Input parameters for the soil erosion model.

| landuse class | PH (m) | NV (stems m$^{-2}$) | D (m) | GC (-) | manning (s m$^{-1/3}$) | sowing (doy)[2] | harvest (doy)[2] | other[1] |
|---|---|---|---|---|---|---|---|---|
| cereal | 0.75 | 500 | 0.025 | 0.31 | n.a. | 288 | 166 | T |
| (harvested) | 0 | 0 | 0 | 0 | n.a. | | | T |
| huerta | 0.5 | 500 | 0.01 | 0.6*CC | n.a. | n.a. | n.a. | T |
| horticulture | 0.3 | 6.25 | 0.25 | 0.39 | n.a. | 288 | 166 | T |
| (harvested) | 0 | 0 | 0 | 0 | n.a. | | | T |
| tree crops | 2 | n.a. | n.a. | <0.01 | n.a. | n.a. | n.a. | T, N.V. |
| vineyard | 1 | n.a. | n.a. | 0.02 | n.a. | n.a. | n.a. | T, N.V. |
| forest | 10 | n.a. | n.a. | 0.57*CC | 0.2[3] | n.a. | n.a. | |
| shrubland | 0.5 | n.a. | n.a. | 0.8*CC | 0.1[3] | n.a. | n.a. | |
| water/urban | 0 | 0 | 0 | 0 | n.a. | n.a. | n.a. | N.E. |

[1] T = tillage, N.E. = no erosion, N.V. = no vegetation, [2] Day of the Year, [3] Obtained from Chow (1959)

cover fraction ($GC$) and, optionally, the Manning's roughness coefficient for vegetation ($n_{\text{vegetation}}$). The user needs to specify whether the landuse class is non-erodible (e.g. pavement and water), tilled and non-vegetated (e.g. bare soil or tilled orchards). We obtained values for each of these parameters through observations from aerial photographs, expert judgement and as part of the calibration procedure. The tillage parameter $RFR$ was set to 6, which corresponds to Cultivator tillage from (Table IV; Morgan and Duzant, 2008). The input parameters change when a crop is harvested, therefore, we varied the input parameters according to the sowing-harvest cycle representing the cropping cycle for horticulture and cereals. Table 1 shows the values of all the landuse-specific input parameters after calibration.

We applied the dynamic vegetation module to obtain crop coefficients and vegetation cover from NDVI, which we obtained from the 16-day Moderate Resolution Imaging Spectroradiometer (MODIS; Didan, 2015) data for the period 2000-2012. For model calibration (2001-2010) we used each of the individual NDVI images, after gap-filling (mainly due to cloud cover) with the long-term average 16-day period NDVI for the period 2000-2012.

For the model validation period (1981-2000) no NDVI images of sufficient quality and resolution were available, therefore we prepared NDVI model input accounting for the intra- and inter-annual variability. The intra-annual variability was obtained from the long-term average 16-day period NDVI for the period 2000-2012. The inter-annual variability was determined based on a log-linear relationship between the annual precipitation sum, annual mean temperature, annual maximum temperature and yearly-averaged NDVI for each of the 25 landuse classes for the period 2000-2012:

$$
\begin{aligned}
NDVI_{\text{year}} = {} & \beta_0 + \log(P_{\text{year}})\beta_1 + \log(P_{\text{year-1}})\beta_2 + \log(Tavg_{\text{year}})\beta_3 \\
& + \log(Tavg_{\text{year-1}})\beta_4 + \log(Tmax_{\text{year}})\beta_5 + \log(Tmax_{\text{year-1}})\beta_6
\end{aligned}
\tag{31}
$$



Where $NDVI$ is the yearly-averaged NDVI, $P$ the annual precipitation sum, $Tavg$ the annual mean temperature, $Tmax$ the annual maximum temperature, and $\beta_{0-6}$ coefficients of the log-linear model. We used the annual climate indices of two years, the current year and the previous year, to account for the climate lag that may influence the vegetation development. A stepwise model selection procedure was applied for each of the 25 landuse classes, selecting the best combination of variables

from equation 31 with the lowest AIC (Akaike Information Criterion) in R (version 3.4.0), using the stepAIC algorithm from the MASS package (Venables and Ripley, 2002).

## 3.2    Model Calibration & validation

We calibrated the model for the period 2001-2010. To prevent overfitting and achieve most realistic model calibration we set most of the potential calibration parameters at literature values and maintained the other parameters within reasonable physical

limits of the parameter domain. We used daily discharge time series from the Segura River Basin Agency (Confederación Hidrográfica del Segura) for the Fuensanta reservoir (Figure 4b) to determine model performance for the hydrological model. The calibration procedure consisted of two steps. First, we optimized the water balance by comparing the observed and simulated discharge sum at the Fuensanta discharge station. We adjusted the calibration parameter $\lambda$ from equation 6 to obtain a surface runoff ratio between 2-10%, which previous studies reported to be representative for catchments with similar condi-

tions (Descroix et al., 2001; Mekki et al., 2006; Love et al., 2010; Reaney et al., 2014). We used parameters from the dynamic vegetation module and soil hydraulic properties to optimize the percent bias of the discharge. In the second step, using the parameter set from the first step, we optimized the Nash-Sutcliffe model efficiency (NSE, Nash and Sutcliffe (1970)) at the Fuensanta discharge station by calibrating the routing parameter ($kx$). The calibration resulted in a Nash-Sutcliffe efficiency (NSE) of 0.47 for the daily discharge data, a NSE of 0.76 for the monthly discharge data and a percent bias of 2.3% 5a. Model

validation for the period 1981-2000 resulted in a NSE of 0.25 for the daily discharge data, a NSE of 0.39 for the monthly discharge data and a percent bias of -18.7% 5b.

     To calibrate the soil erosion model we first optimized the detached material going into transport $G$ for 8 landuse classes (aggregated from 25 landuse classes), based on literature data (Cerdan et al., 2010; Maetens et al., 2012) (Table 2). The resulting optimized parameter values are presented in Table 1. Next, we optimized percent bias in prediction of sediment yield

at the reservoirs using measured reservoir sediment yield data from 4 reservoirs (Avendaño-Salas et al., 1997) (see Figure 4b). The calibration procedure focused on the most sensitive parameter from the sediment transport capacity equation 28, i.e. the $\beta$ parameter. We obtained a percent bias of 0.0% in the calibration and -19.8% in the validation (see Figure 6).

## 3.3    Results

Here we present a selection of model results to illustrate the main capabilities of the SPHY-MMF model. Soil erosion shows

an important intra-annual variability due to seasonal changes in climate forcing and vegetation cover (Figure 7). For crops with little to no ground cover (i.e. tree crops and vineyard), soil erosion follows the precipitation sum, with high values in the winter, spring and autumn months and low values in the summer months. Some crops show a distinct peak in the vegetation



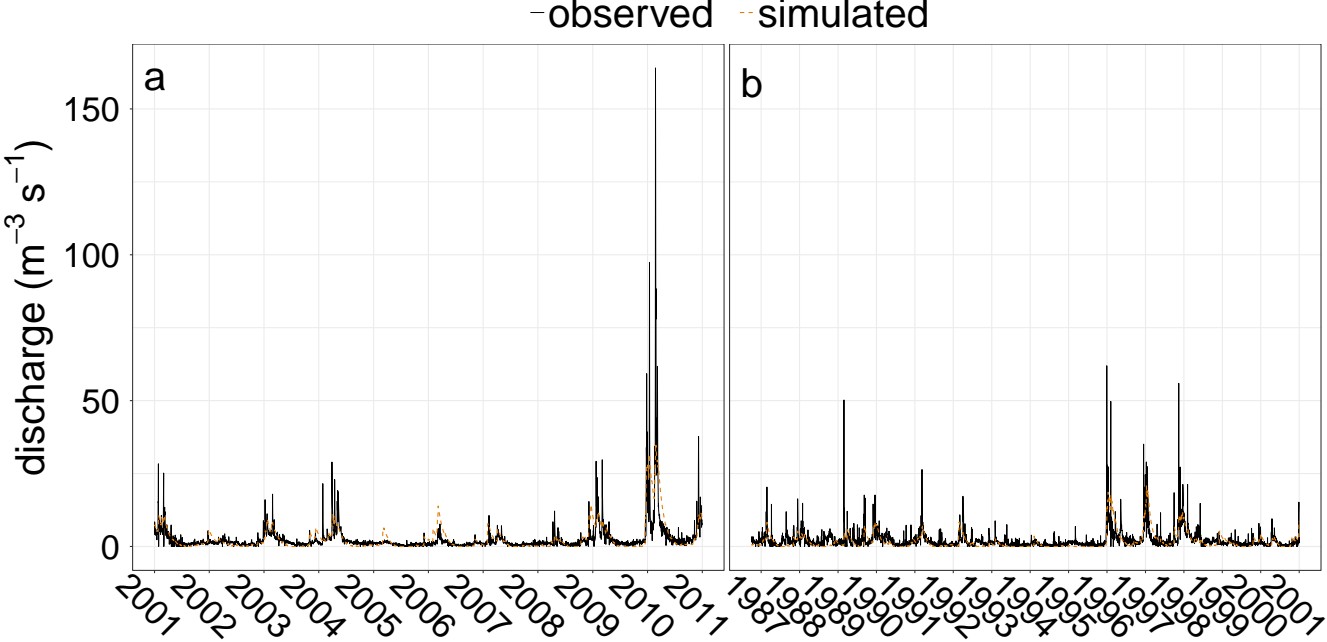

**Figure 5.** Discharge time series for the calibration (a) and validation period (b). The solid line correspond to the observed time series and the dashed orange line corresponds to the simulated time series.

**Table 2.** Calibration and validation of hillslope soil erosion and comparison with literature data ($Mg\,km^{-2}\,yr^{-1}$).

| landuse class | calibration | validation | Cerdan et al. (2010) | Maetens et al. (2012) |
|---|---|---|---|---|
| cereals | 99.8 | 99.2 | 84.0 | 120.0 |
| huerta | 112.0 | 112.6 | 84.0 | 120.0 |
| horticulture | 145.5 | 165.4 | 84.0 | 120.0 |
| tree crops | 249.9 | 236.4 | 167.0 | 740.0 |
| vineyard | 194.7 | 194.6 | 862.0 | 30.0 |
| forest | 17.2 | 13.5 | 18.0 | 10.0 |
| shrubland | 54.9 | 39.1 | 54.0 | 20.0 |
| water/urban | 0.0 | 0.0 | n.a. | n.a. |

development in the spring (April-May), e.g. huerta and horticulture. While this period has a relatively high precipitation sum, soil erosion decreases as a consequence of the increased vegetation cover indicated by the NDVI in this period.

The temporal variation of the vegetation development of cereals and horticulture shows a slightly distinct pattern from the other landuse classes. Both lines show an increase in the spring months (March-May), which indicates the rapid growth of these crops. While during the summer months (June-August) the NDVI decreases, which coincides with the period when the



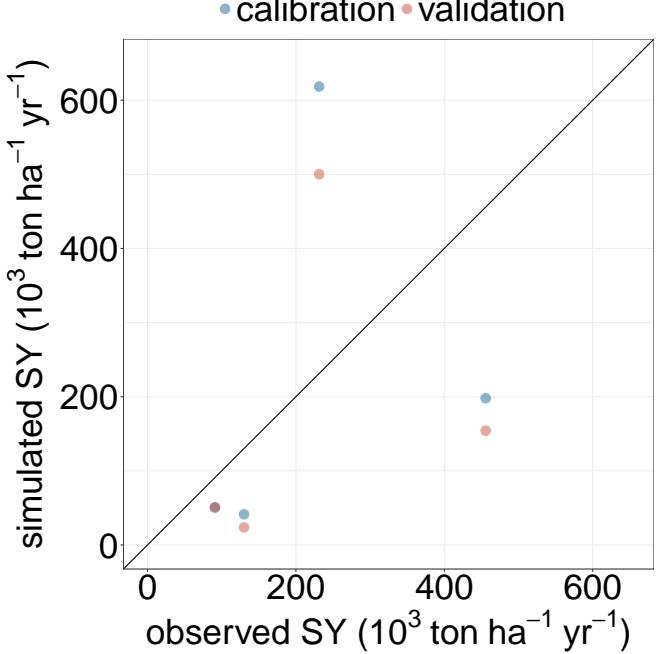

**Figure 6.** Average yearly sediment yield at the reservoirs for the calibration (blue) and validation period (red).

crops are harvested, followed by the post-harvest period. In the latter period, we assume bare soil conditions for these crops. For both crops this ultimately results in the highest annual erosion rates in the post-harvest period (October).

To illustrate the models capacity to perform scenario studies we evaluated the impacts of the application of sustainable land management and the impacts of a future climate change scenario. First, to assess the impacts of sustainable land management,

we evaluated the application of conservation agriculture by assuming that no bare soil conditions occur after harvest of cereals and horticulture for the period June-October (dashed lines in Figure 7). This leads to lower soil erosion estimates for these two landuse classes, as a result of an increase of immediate deposition of detached particles.

Next, we simulated the impacts of a projected climate change scenario, by comparing predicted soil erosion rates and sediment yield under the reference scenario (1981-2000) with a future scenario (2081-2100). We used a future emission scenario

from the Representative Concentration Pathways (RCP; van Vuuren et al., 2011) that describes a continuous increase of GHG emissions throughout the 21[st] century, i.e. RCP8.5. For this exercise we used projected climate data for RCP8.5 obtained from one Regional Climate Model (CLMcom MPI-ESM-LR) from the EURO-CORDEX initiative (Jacob et al., 2014), for the period 2081-2100. The climate forcing (precipitation and temperature) was bias-corrected using quantile mapping (Themeßl et al., 2012) and we applied the dynamic vegetation model (Equation 31) to construct future NDVI input based on future climate

conditions. Figure 8 shows the precipitation and vegetation response under the reference and future scenarios. The annual precipitation sum decreases in the future scenario, with a catchment-averaged decrease of 128 $\mathrm{mm}$. However, heavy precipitation, defined as the 95th percentile of daily precipitation, considering only rainy days ($>1\,\mathrm{mm\,day^{-1}}$; Jacob et al., 2014), increases





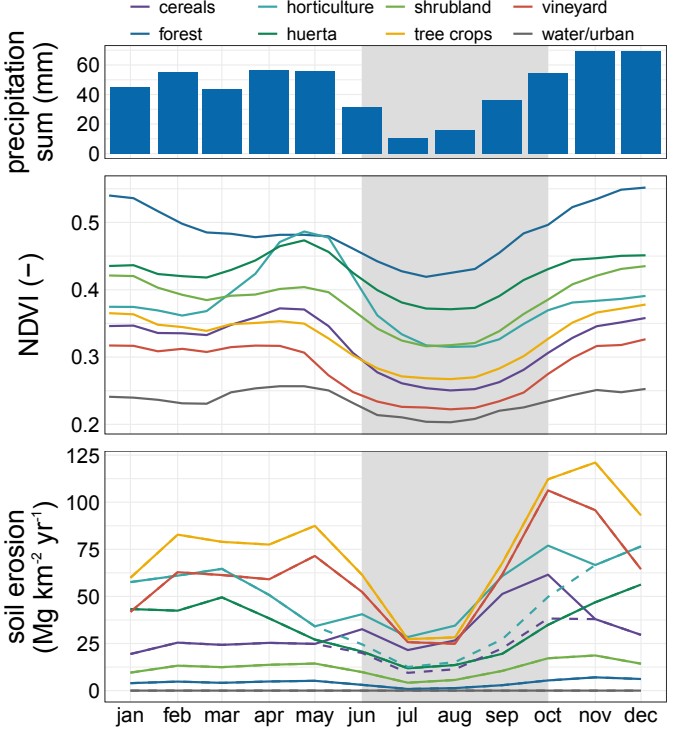

**Figure 7.** Monthly precipitation sum $(\mathrm{mm})$, NDVI (-) and soil erosion $(\mathrm{Mg\,km^{-2}\,yr^{-1}})$ per landuse class for the period 1981-2000. The gray area indicates the period when cereals and horticulture are harvested and model parameters are changed to simulate bare soil conditions. The dashed lines in the lower panel show the soil erosion of cereals and horticulture without considering bare soil conditions in this period.

by 27% on average in the catchment. The NDVI increases in the western part of the catchment due to increasing temperatures and decreases in the eastern part of the catchment due to a combination of decreasing precipitation and increasing temperatures.

In the reference scenario the highest specific sediment yield (SSY) is projected in the river network (Figure 9), where accumulated runoff causes an increase of soil erosion rates (Equation 19). In the future climate scenario, the catchment-median

5  SSY increases from 43.3 to $55.2\,\mathrm{Mg\,km^{-2}\,yr^{-1}}$, an increase of 27.7%. This shows that the increase in heavy precipitation has a more pronounced impact on soil erosion than the decrease of annual precipitation sum. The increase of heavy precipitation both leads to an increase of detachment by raindrop impact (Equation 18) and an increase of detachment by runoff (Equation 19), as a consequence of an increase in surface runoff due to infiltration excess surface runoff. However, as a result of the increased vegetation cover in the western part of the catchment SSY decreases, despite the increased extreme precipitation

10  intensities (Figure 8).

Reservoir sediment yield (SY) decreases in all five reservoirs between 42.4-59.0% in the future climate scenario. While it is likely that a decrease of SSY in the western part of the catchment causes a decrease of reservoir SY, it is less obvious why in the eastern part of the catchment an increase in SSY is not reflected in an increase in reservoir SY. The explanation for this





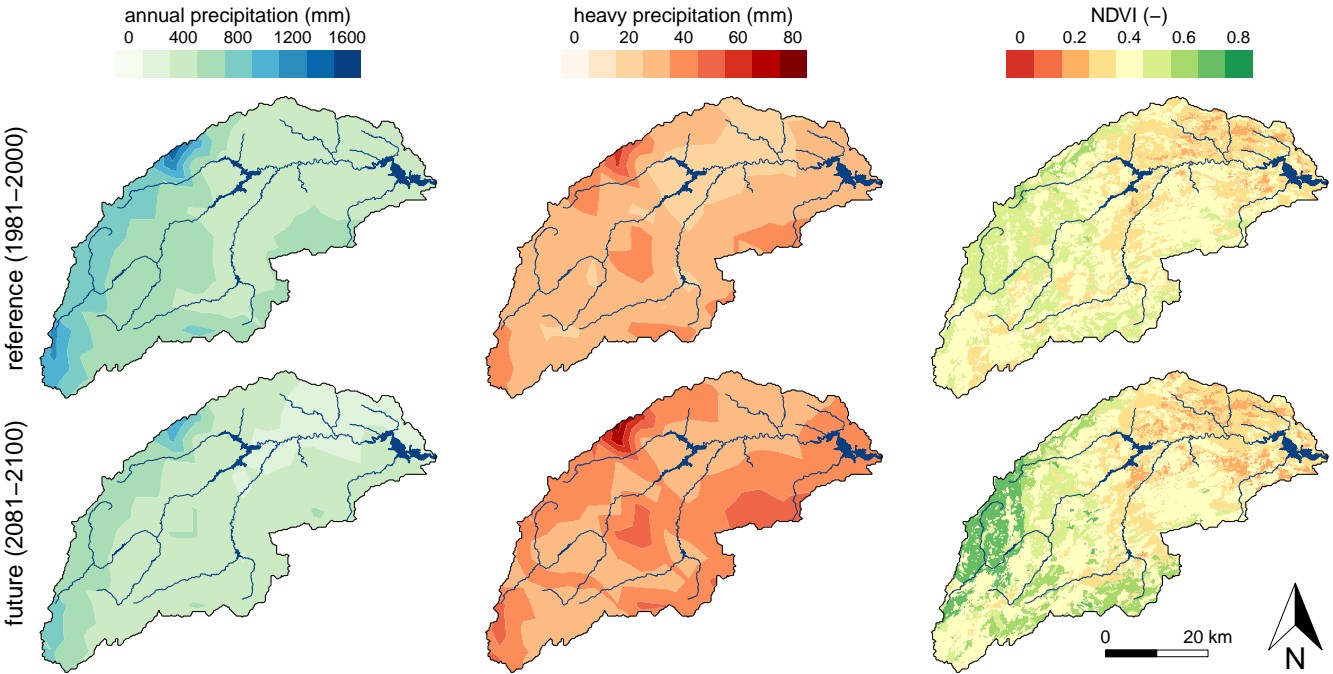

**Figure 8.** Average annual precipitation sum (mm), heavy precipitation (mm) and NDVI (-) for the reference (1981-2000) and future (2081-2100) scenarios.

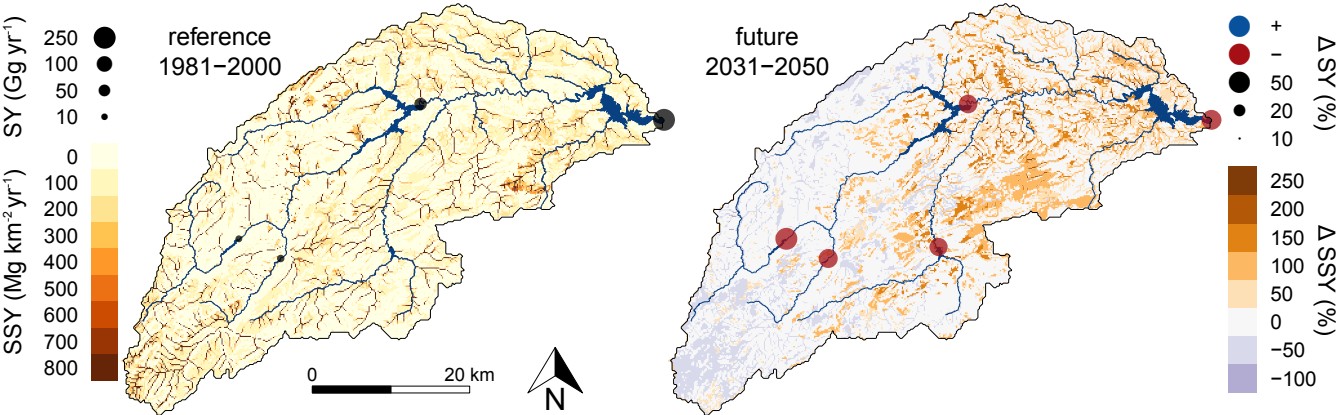

**Figure 9.** Specific sediment yield ($\mathrm{Mg\,km^{-2}\,yr^{-1}}$) and reservoir sediment yield ($\mathrm{Gg\,yr^{-1}}$) for the reference (1981-2000) scenario and the change (%) for the future (2081-2100) scenario.

lies in the fact that a decrease in precipitation sum causes a decrease of accumulated runoff and, subsequently, a decrease of sediment transport capacity (Equation 28), increased sediment deposition and decreased reservoir SY.



## 4  Discussion

The SPHY-MMF model, based on integration of the SPHY hydrological model with the MMF soil erosion model, provides an important step forward to simulate the regional scale impacts of environmental change on soil erosion and sediment yield. The model runs at a daily time step, incorporates the main hydrological driving processes (i.e. saturation and infiltration excess surface runoff), accounts for the most relevant soil erosion processes (i.e. soil erosion by raindrop impact, soil erosion by accumulated runoff and sediment deposition) and incorporates a dynamic vegetation module that is linked to both the hydrological and the soil erosion model. This provides the model with flexibility and accuracy needed to reflect the impacts of intra-annual changes in land use, land management and climate (Figure 7) and the inter-annual response of vegetation development and soil erosion to changes in climate forcing, including changes in precipitation sum and intensity (Figure 9).

Availability of high quality input data is an important constraint for many process-based erosion models. Although SPHY-MMF requires a wide range of input data, most data were obtained from publicly available global datasets at relatively high spatial and temporal resolution. Although climate data are often still an important constraint, long-term, high-resolution, gridded daily climate forcing datasets are becoming increasingly more available at national (Silva et al., 2007; Yin et al., 2015; Berezowski et al., 2016; Kotlarski et al., 2017), (sub-)continental (Mitchell, 2004; Haylock et al., 2008; Yatagai et al., 2012; van den Besselaar et al., 2017) and even at global scale Huffman et al. (2001); Donat et al. (2013); Schamm et al. (2016). The model also requires a detailed landuse map to account for landuse-specific model parameters. We obtained literature values for two model parameters, i.e. depletion fraction and the maximum LAI, while other model parameters were obtained through observations from aerial photographs, expert judgement and as part of the calibration procedure. This may be inevitable because soil erosion is inherently landuse dependent and model assessments should involve the inclusion of detailed local landuse data to get reliable results. However, most other input datasets are publicly available, which makes the model applicable for any environment.

The model results show that the highest soil erosion rates are projected where most flow accumulates (Figure 9). This is mainly due to the high amounts of soil erosion predicted by Equation 19 for large runoff volumes. Similar behaviour was reported in a daily implementation of MMF model by (Shrestha and Jetten, 2018), who therefore suggested to exclude higher order streams from the soil erosion assessment. We correct for the high erosion rates in the river network by including both immediate sediment deposition (Equation 20) and deposition when the transport capacity is exceeded (Equation 28). While indeed different erosional processes dominate in channels and streams (i.e. bank erosion and channel incision), which are not captured by Equation 19, high soil erosion rates may be expected in the river network since many studies stress the large contribution of gully and channel erosion to total sediment yield due to large volumes of accumulated runoff (Poesen et al., 1996; Poesen, 2018). However, detachment and transport processes in channels are likely different from those on hillslopes and the bed material of rivers differs from the soil texture. While Equation 19 makes a distinction between the three textural classes, it most likely overestimates the erosion in the higher order channels, where the bed material consists of coarser material, such as coarse sand and gravel. Therefore, in a future update of the model, we suggest to include a separate channel erosion,





transport and deposition model, which should be able to simulate the most relevant channel erosion and deposition processes more accurately.

## 5   Conclusions

We have presented a new coupled hydrology, soil erosion and sediment yield prediction model (SPHY-MMF), and its appli-
cation to the Upper Segura catchment for different climate and land management scenarios. The model is an integration of the MMF soil erosion model in the SPHY hydrological model and simulates most relevant hydrological and soil erosion pro-
cesses at a daily time-step. The model considers soil detachment by raindrop and runoff, uses dynamic vegetation to simulate changes in the canopy cover, simulates saturation excess and infiltration excess surface runoff, simulates soil deposition in the cell of its origin and routes the sediment through the river network, considering the transport capacity of the flow. The model
was successfully applied in a large catchment in southeastern Spain. We have shown that the model is capable of performing scenario assessments of changes in climate and land management. Furthermore, our results show that the model simulates the soil erosion response to intra-annual variability in climate conditions and vegetation development. While there remain multiple challenges to accurately simulating the impacts of environmental change on soil erosion and sediment yield, we consider the integrated SPHY-MMF model an important step forward to facilitate catchment scale scenario studies.

*Code availability.*   The model source code is available online: https://github.com/JorisEekhout/SPHY/tree/SPHY2.1-MMF.

*Competing interests.*   The author declares that they have no conflict of interest.

*Acknowledgements.*   We acknowledge financial support from the "Juan de la Cierva" program of the Spanish Ministerio de Economía y Competitividad (FJCI-2016-28905), the Spanish Ministerio de Economía y Competitividad (ADAPT project; CGL2013-42009-R) and the Séneca foundation of the regional government of Murcia (CAMBIO project; 118933/JLI/13). The authors thank AEMET and UC for the
data provided for this work (Spain02 v5 dataset, available at http://www.meteo.unican.es/datasets/spain02).



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
