# Peer review of "Assessing the large-scale impacts of environmental change using a coupled hydrology and soil erosion model"

_Earth Surface Dynamics, 2018_

## Referee Comment (RC1) · A.J.A.M. Temme (Referee) · 7 May 2018

The paper presents an innovative combination of the existing SPHY hydrological and MMF erosion models, with intended use at the large catchment and decadal scales. Despite this foreseen use at large spatial and temporal scales, the combined model is presented as a physically-based model. I agree with this assessment: most relevant processes that determine catchment water and sediment yield are included, and well presented in the paper. My main concern in this part of the paper is that there is insufficient attention for the large number of parameters that these extra formulae introduce. I recommend that the reader be provided with more information about all parameters,

especially whether they are 1) measurable or not, 2) available from literature, and if so for which kind of environments, or 3) most be obtained through calibration. Ideally, this would go together with a detailed global sensitivity analysis, that could give insight in how robust model simulations are under the weight of all these parameters. A good example for how to do this is https://www.geosci-model-dev-discuss.net/gmd-2017-236/ , or (I apologize) my own paper in Computers and Geosciences about the Lorica model a few years back. If the authors find that sensitivity analysis is a bridge too far in this already long paper, then that lends more weight to my first suggestion (provide details - even if it happens in supplementary material).

The model is then calibrated and validated to simulate water and sediment yield for a large catchment in Spain, and two scenarios for the future state of the catchment are used to demonstrate the models' capability. The quality of calibration and validation results is not concerning at first sight, but should be placed in the context of other models' results - even though these are bound to be in other areas, for other timescales, etc. Only reporting such results leaves the reader with questions. During my review, I also missed reporting on results from the first (conservation agriculture) scenario - a climate change scenario appeared to get all the attention.

Figures in the paper are good as it stands - although a bit more detail in them and their captions would make them more useful. I would appreciate another table with information on parameters (see my comment above). Detailed annotations and suggestions are available in the attached scan.

All in all, I consider this an important contribution to literature presenting a model that usefully supplements the range of available models. I especially appreciate the authors' detailed and deliberate explanation of model equations. In light of the modest suggestions for improvement, I suggest minor revisions.

Arnaud Temme

Please also note the supplement to this comment:
https://www.earth-surf-dynam-discuss.net/esurf-2018-25/esurf-2018-25-RC1-supplement.pdf

**ESurfD**
[Figure]

**Supplement:**

[revised manuscript text omitted]

→ By now you should become a bit more specific about
which models you are thinking of. Perhaps a table
works best.

[Figure]

*[handwritten: larger]*

*[handwritten: just say 'empirical' unless you are challenging that.]*

At these scales, soil erosion is often assessed using so-called empirical erosion models. These models are derived from field studies where soil erosion has been observed under different land use, management, soil, climate, and topographical conditions. The most well-known and applied *[handwritten: best]* empirical model is the Universal Soil Loss Equation (USLE) Wischmeier and Smith (1978) and its derivatives RULSE Renard et al. (1997) and MUSLE Williams (1995). While the empirical formulations of the USLE

5   were obtained at plot-scale, the model is often applied at much larger scales, sometimes in combination with a sediment *[handwritten: (refs)]* transport capacity equation or a sediment delivery ratio to assess sediment yield. Due to its simplicity, the USLE can be applied with a relatively limited amount of input data. However, their main restriction is the limited number of processes accounted for (e.g. the USLE and RUSLE based models only consider sheet and rill erosion) and the limited potential to evaluate the impacts of changes in climate and land management. Furthermore, these models are typically applied at annual time steps,

10   largely neglecting intra-annual variation of climate and vegetation conditions.

Most current soil erosion models have a limited potential for application at larger temporal and spatial scales (i.e. process-based models) or lack sufficient representation of the underlying soil detachment and sediment transport processes and sensitivity to changes in land use or climate (i.e. empirical models), making them of limited use for scenario studies and process understanding. Here, we present a process-based soil erosion model based on the integration of the Morgan-Morgan-Finney

15   erosion model (MMF; Morgan and Duzant, 2008) and the spatially distributed hydrological model Spatial Processes in HYdrology (SPHY; Terink et al., 2015). This integrated model overcomes many of the limitations of previous large-scale soil erosion models, as it includes a more complete representation of crucial processes like surface runoff generation, dynamic vegetation development, and sediment deposition, and runs at the catchment scale with a daily time step. This makes the model especially suitable for evaluation of the inter- and intra-annual impacts of environmental change on soil erosion and sediment yield at

20   large spatial scales. In the next paragraphs we first present the different model components and enhancements as compared to previous models. Then we illustrate its functionality and potential for scenario studies by application to the Upper Segura catchment in southeastern Spain under present and projected future climate conditions.

*[handwritten right margin: there are so many references to choose from, that it should not lead to you not showing any.]*

**2   Model Description**

**2.1   Model Overview**

*[handwritten: such as]*

*[handwritten: Introduction: well structured, clear, but insufficiently supported by references!]*

25   The SPHY-MMF model presented here is an integration of the (Modified) Morgan-Morgan-Finney soil erosion model into the SPHY hydrological model (version 2.1). Figure 1 shows the main hydrological and soil erosion processes considered by the model. SPHY is a spatially distributed leaky-bucket type model that simulates hydrological processes on a cell-by-cell basis at a daily timestep (Terink et al., 2015). The model is written in the Python programming language using the PCRaster dynamic modelling framework (Karssenberg et al., 2010). MMF is a conceptual soil erosion model that originally is applied with an

30   annual time step. Here we present a modification of the model at a daily time step, fully integrated with the SPHY model. MMF receives input from the SPHY model, such as effective precipitation (throughfall), runoff and canopy cover for calculation of erosion and deposition processes.

*[handwritten: Especially at daily resolution, should the erosion model not feed back to SPHY as well? You erode topsoil, with implications for infiltration and erodibility. + deposit]*

[Figure]

[Figure]

**Figure 1.** Overview of the model: (a) representation of a single cell, (b) the hydrological processes, and (c) the soil erosion processes.

*Caption needs to explain the acronyms in C.*

**2.2 Hydrological model**

SPHY simulates most relevant hydrological processes (Figure 1b), such as interception, evapotranspiration, dynamic evolution of vegetation cover, surface runoff, and lateral and vertical soil moisture flow at a daily time step. The model is described in full detail by Terink et al. (2015), therefore, here we only provide a summary of the processes that are simulated by the model,

5    some hydrological processes that have been changed with respect to the original SPHY model, and a detailed description of the processes that are related to the integration of MMF.

SPHY requires daily precipitation and temperature maps as input. Effective precipitation is determined by subtracting canopy storage and interception from precipitation. Canopy storage is determined from the Leaf Area Index (LAI), which is derived from Normalized Differenced Vegetation Index (NDVI) images. Reference evapotranspiration is determined using the Harg-

10   reaves equation (Hargreaves and Samani, 1985), which is subsequently multiplied by the crop coefficient to obtain the potential evapotranspiration. The crop coefficient is determined from the NDVI images using a linear relationship. Actual evapotranspiration is obtained by multiplying the potential evapotranspiration by a reduction factor for water deficit or water surplus, which are functions of current soil water content, soil hydraulic properties and plant-specific water need. Surface runoff is determined by a daily implementation of the Green-Ampt formula and is a function of infiltration, effective precipitation and soil hydraulic

15   properties. The soil profile consists of three layers, i.e. rootzone, subzone and groundwater layer. Water can percolate from

*reference*

[Figure]

*does SFA limit your model's usefulness to non-ploughed / i.e. non smoothed landscapes?*

the rootzone to the subzone and from the subzone to the groundwater layer. Water travels from the subzone to the rootzone through capillary rise. Water drains from the rootzone as lateral flow and from the groundwater layer as baseflow. The total runoff is the sum of surface runoff, lateral flow and baseflow. All soil processes are functions of current water content (in the respective layers) and soil hydraulic properties, i.e. saturated hydraulic conductivity, saturated water content, field capacity and

5 wilting point. Water is routed using a single flow algorithm. A flow recession coefficient accounts for flow delay from channel friction. When reservoirs are present, the user can opt to include an advanced routing scheme accounting for reservoir storage and outflow.

*I am very sympathetic of brief model descriptions, so I like 2.2. However, it gives me the idea that this 'physical-based model' still has many calibration parameters. Can you give us an overview of measurable / calibration parameters*

[revised manuscript text omitted]

*[handwritten annotation: does that not follow from Eqs. 20-22 and the amount of water in the reservoir?]*

The amount of sediment that is routed to downstream cells is the summation of the individual amounts for clay, silt and sand.

10 Sediment is routed using a routing scheme that takes into account both the transport capacity ($TC$; ton ha$^{-1}$) of the accumulated runoff and the trapping efficiency of the reservoirs ($TE$; -). The transport capacity $TC$ (Figure 3a) of the accumulated runoff is based on Prosser and Rustomji (2000):

$$TC = \text{flow}_{\text{factor}} q_{surf}^{\beta} S^{\gamma} \tag{28}$$

Where $\text{flow}_{\text{factor}}$ is a spatially distributed roughness factor (-), $q_{surf}$ accumulated runoff per unit width (m$^2$ day$^{-1}$), $S$ the local

15 energy gradient (°), approximated by the slope, and $\beta$ and $\gamma$ are model parameters (-). As suggested by Prosser and Rustomji (2000) $\gamma = 1.4$ and $\beta$ is used for model calibration.

The roughness factor $\text{flow}_{\text{factor}}$ is determined as follows:

$$\text{flow}_{\text{factor}} = \frac{v_{\text{actual}}}{v_{\text{b}}} \tag{29}$$

Where $v_{\text{actual}}$ is the actual flow velocity (m s$^{-1}$) and $v_{\text{b}}$ is the flow velocity for bare soil conditions (m s$^{-1}$). The actual flow

20 velocity $v_{\text{actual}}$ is obtained from Equations 23-26, applying a water depth $d$ of 0.25 m, which coincides with deeper rills from Morgan and Duzant (2008). The flow velocity for bare soil conditions $v_{\text{b}}$ is obtained from Equation 23, applying values for $n' = 0.015$ s m$^{-1/3}$ and $d = 0.005$ m (Morgan and Duzant, 2008).

Reservoir sediment trapping efficiency $TE$ (Figure 3b), the percentage of sediment trapped by the reservoir, is calculated according to Brown (1943):

25 $$TE = 100 \left[ 1 - \frac{1}{1 + 0.0021 D \frac{C}{A_{\text{basin}}}} \right] \tag{30}$$

Where $D$ is a constant (-) within the range 0.046-1, depending on the reservoir operation characteristics that we set at 0.1, $C$ the reservoir capacity (m$^3$), and $A_{\text{basin}}$ the drainage area of the subcatchment (km$^2$).

Earth **Surface**
Dynamics
Discussions

*With Ch. 2 in mind:*
*① Great and calm explanation. I think*
*I know what this model does.*
*② The model has many parameters. A*
*table, perhaps in suppl. mat. would help*
*to distinguish measurable, literature*
*based and calibrated*
*parameters*

[Figure]

*like this*
*but nicer*

**Figure 3.** Sediment routing: (a) transport of sediment through the catchment, and (b) trapping efficiency at the reservoirs.

*These figures are great but can be made more*
*useful by referring to your equations in them.*

**3 Model Application**

[revised manuscript text omitted]

*You do not discuss the conservation agriculture scenario results. Or is that your 'reference' scenario? That would be confusing. and should be improved.*

Earth **Surface**
Dynamics
Discussions

[Figure]

*I like figs 8+9 very much, but I miss the findings of the sustainable land man scen*

[Figure]

**Figure 8.** Average annual precipitation sum (mm), heavy precipitation (mm) and NDVI (-) for the reference (1981-2000) and future (2081-2100) scenarios.

[Figure]

**Figure 9.** Specific sediment yield ($\mathrm{Mg\,km^{-2}\,yr^{-1}}$) and reservoir sediment yield ($\mathrm{Gg\,yr^{-1}}$) for the reference (1981-2000) scenario and the change (%) for the future (2081-2100) scenario.

lies in the fact that a decrease in precipitation sum causes a decrease of accumulated runoff and, subsequently, a decrease of sediment transport capacity (Equation 28), increased sediment deposition and decreased reservoir SY.

[Figure]

*this feels a little too much like an abstract or an advertisement. better at the end if in discussion, after you have provided the reader with the insights that support it.*

**4  Discussion**

The SPHY-MMF model, based on integration of the SPHY hydrological model with the MMF soil erosion model, provides an important step forward to simulate the regional scale impacts of environmental change on soil erosion and sediment yield. The model runs at a daily time step, incorporates the main hydrological driving processes (i.e. saturation and infiltration

5  excess surface runoff), accounts for the most relevant soil erosion processes (i.e. soil erosion by raindrop impact, soil erosion by accumulated runoff and sediment deposition) and incorporates a dynamic vegetation module that is linked to both the hydrological and the soil erosion model. This provides the model with flexibility and accuracy needed to reflect the impacts of intra-annual changes in land use, land management and climate (Figure 7) and the inter-annual response of vegetation development and soil erosion to changes in climate forcing, including changes in precipitation sum and intensity (Figure 9).

10  Availability of high quality input data is an important constraint for many process-based erosion models. Although SPHY-MMF requires a wide range of input data, most data were obtained from publicly available global datasets at relatively high spatial and temporal resolution. Although climate data are often still an important constraint, long-term, high-resolution, gridded daily climate forcing datasets are becoming increasingly more available at national (Silva et al., 2007; Yin et al., 2015; Berezowski et al., 2016; Kotlarski et al., 2017), (sub-)continental (Mitchell, 2004; Haylock et al., 2008; Yatagai et al., 2012;

15  van den Besselaar et al., 2017) and even at global scale Huffman et al. (2001); Donat et al. (2013); Schamm et al. (2016). The model also requires a detailed landuse map to account for landuse-specific model parameters. We obtained literature values for two model parameters, i.e. depletion fraction and the maximum LAI, while other model parameters were obtained through observations from aerial photographs, expert judgement and as part of the calibration procedure. This may be inevitable because soil erosion is inherently landuse dependent and model assessments should involve the inclusion of detailed local landuse data

20  to get reliable results. However, most other input datasets are publicly available, which makes the model applicable for any environment.

*aren't they really there?*

The model results show that the highest soil erosion rates are projected where most flow accumulates (Figure 9). This is mainly due to the high amounts of soil erosion predicted by Equation 19 for large runoff volumes. Similar behaviour was reported in a daily implementation of MMF model by (Shrestha and Jetten, 2018), who therefore suggested to exclude higher

25  order streams from the soil erosion assessment. We correct for the high erosion rates in the river network by including both immediate sediment deposition (Equation 20) and deposition when the transport capacity is exceeded (Equation 28). While indeed different erosional processes dominate in channels and streams (i.e. bank erosion and channel incision), which are not captured by Equation 19, high soil erosion rates may be expected in the river network since many studies stress the large contribution of gully and channel erosion to total sediment yield due to large volumes of accumulated runoff (Poesen et al.,

30  1996; Poesen, 2018). However, detachment and transport processes in channels are likely different from those on hillslopes and the bed material of rivers differs from the soil texture. While Equation 19 makes a distinction between the three textural classes, it most likely overestimates the erosion in the higher order channels, where the bed material consists of coarser material, such as coarse sand and gravel. Therefore, in a future update of the model, we suggest to include a separate channel erosion,

*if that's your main finding, a much simpler model would do. Ample literature confirms this finding.*

*well.... you calibrated a few other params, and literature values are not always directly transferable. This aspect deserves more attention.*

transport and deposition model, which should be able to simulate the most relevant channel erosion and deposition processes more accurately.

*Show/cite some examples from literature. People must have done this for a long time, I think*

**5 Conclusions**

We have presented a new coupled hydrology, soil erosion and sediment yield prediction model (SPHY-MMF), and its application to the Upper Segura catchment for different climate and land management scenarios. The model is an integration of the MMF soil erosion model in the SPHY hydrological model and simulates most relevant hydrological and soil erosion processes at a daily time-step. The model considers soil detachment by raindrop and runoff, uses dynamic vegetation to simulate changes in the canopy cover, simulates saturation excess and infiltration excess surface runoff, simulates soil deposition in the cell of its origin and routes the sediment through the river network, considering the transport capacity of the flow. The model was successfully applied in a large catchment in southeastern Spain. We have shown that the model is capable of performing scenario assessments of changes in climate and land management. Furthermore, our results show that the model simulates the soil erosion response to intra-annual variability in climate conditions and vegetation development. While there remain multiple challenges to accurately simulating the impacts of environmental change on soil erosion and sediment yield, we consider the integrated SPHY-MMF model an important step forward to facilitate catchment scale scenario studies.

*Code availability.* The model source code is available online: https://github.com/JorisEekhout/SPHY/tree/SPHY2.1-MMF.

*Competing interests.* The author declares that they have no conflict of interest.

*Acknowledgements.* We acknowledge financial support from the "Juan de la Cierva" program of the Spanish Ministerio de Economía y Competitividad (FJCI-2016-28905), the Spanish Ministerio de Economía y Competitividad (ADAPT project; CGL2013-42009-R) and the Séneca foundation of the regional government of Murcia (CAMBIO project; 118933/JLI/13). The authors thank AEMET and UC for the data provided for this work (Spain02 v5 dataset, available at http://www.meteo.unican.es/datasets/spain02).

---

## Referee Comment (RC2) · A. Millares (Referee) · 3 Jun 2018

First of all, I would like to congratulate the authors for the effort and dedication developed in this work. Any proposal of erosion modeling that goes beyond the estimation of potential annual erosive rates is an appreciable step forward for the scientific community, managers and, hence, the society. Reading the document has been easy, interesting and instructive to me. My general concerns are the following

1) Studying the configuration of the model, with a clearly hillslope nature, I see its particular strength more related to the temporal scale of simulation, not so much with the spatial scale, and also with the role of vegetation on sediment transport and erosion

processes. That is, both the considered $dt$ and the adopted spatial resolution with its consequent limitations for the modeling of processes in Mediterranean environments can be justified by the model's ability to generate long time series based on processes and distributed information under different land use or climate scenarios. This is not reflected in the current manuscript and, in my opinion, its limitations are then not sufficiently justified.

2) The limitations of the model have to be declared and analyzed more in depth. Especially, I see problematic the modeling of extreme events, which are important in Mediterranean environments with frequent sub-hour time pulses. What is the implication of the runoff model proposed in the results?, as they indicate a considerable increase in this type of events in the considered future scenario . Has sub-hour rainfall data been analyzed? Could the model be modified to include these cases, ($\alpha = 1$ at $t = n$ and $\alpha = 0$ at $t \neq n$?), I see an attenuation effect by the model for this cases. Also, I see inconsistencies, or I do not understand, the units in this part of the model ( $\alpha$ in $h^{-1}$?, Qsurf mm/day?), please clarify. Another limitation is related to the selected cell size (200m grid size), Why this resolution?. Is it related to the computational cost of the model? (I would like to know something about this issue), or is related to the remote sensing images?. What limitations does this present from the point of view of the observed erosion processes in the study area, the forcing agents and their spatial distribution from the obtained results?.

3) Fluvial vs hillslope contributions: the authors declare limitations of the model for modeling fluvial transport, which, in my opinion, should be assessed in the future from a fluvial sub-model that includes the basic erosion and transport processes (bedload + SS), integrating what that comes from the hillslopes (water and sediment) at different points. However, if I'm not wrong, the calibration/validation has been made from SSY estimated from reservoirs, what fraction corresponds to fluvial/hillslope contributions based on reservoir measurements?, this analysis is important and should be adressed given the process-based nature of the model.

4) Although the document is well written and easy to read, the introduction seems too long and could be summarized. On the contrary, more detailed information about the validation, calibration data, especially of the selected reservoirs (bathymetries, topographies, sediment grain sizes, ...) and SSY assessment is missing. It may be interesting to incorporate a section of uncertainty analysis. Please, check the citation format of the entire document.

In general, I encourage the authors to highlight the most interesting aspects of the model, taking into account the spatial and temporal scales adopted, as well as reasonably stating of the associated limitations in order to evaluate not only the model, but all the related challenges. Some minor comments are reported in the attached file.

Please also note the supplement to this comment:
https://www.earth-surf-dynam-discuss.net/esurf-2018-25/esurf-2018-25-RC2-supplement.pdf

**Supplement:**

[revised manuscript text omitted]

---

## Author Comment (AC1) · 5 Jul 2018

We have attached a pdf document that includes the response to the reviews, the revised manuscript with track changes and supplementary information.

Please also note the supplement to this comment:
https://www.earth-surf-dynam-discuss.net/esurf-2018-25/esurf-2018-25-AC1-supplement.pdf

---

## Author Response (AR1)

**Response to reviews of "Assessing the large-scale impacts of environmental change using a coupled hydrology and soil erosion model" submitted to *Earth Surface Dynamics* for consideration for publication.**

*We warmly thank the two reviewers and the editor for their positive and constructive reviews of our manuscript. Below we provide a response to their concerns and explain which revisions were implemented and why a certain approach was taken. All changes are indicated in the document with indication of track changes.*

**Referee #1**

The paper presents an innovative combination of the existing SPHY hydrological and MMF erosion models, with intended use at the large catchment and decadal scales. Despite this foreseen use at large spatial and temporal scales, the combined model is presented as a physically-based model. I agree with this assessment: most relevant processes that determine catchment water and sediment yield are included, and well presented in the paper.

*We would like to thank the reviewer for his nice comments on the manuscript.*

My main concern in this part of the paper is that there is insufficient attention for the large number of parameters that these extra formulae introduce. I recommend that the reader be provided with more information about all parameters, especially whether they are 1) measurable or not, 2) available from literature, and if so for which kind of environments, or 3) most be obtained through calibration. Ideally, this would go together with a detailed global sensitivity analysis, that could give insight in how robust model simulations are under the weight of all these parameters. A good example for how to do this is https://www.geosci-model-dev-discuss.net/gmd-2017-236/, or (I apologize) my own paper in Computers and Geosciences about the Lorica model a few years back. If the authors find that sensitivity analysis is a bridge too far in this already long paper, then that lends more weight to my first suggestion (provide details - even if it happens in supplementary material).

*Indeed, the model contains a large number of parameters, although most of them we obtained from literature and are referenced in the manuscript. Still, we agree that a table that contains a comprehensive list of all the parameters is very useful. Therefore, we have added a table to the Supplementary Material that includes all model parameters mentioned in the manuscript. The table indicates the following aspects: landuse-specific, measurable, calibration, literature based and a citation where we obtained the literature values of the parameters.*

The model is then calibrated and validated to simulate water and sediment yield for a large catchment in Spain, and two scenarios for the future state of the catchment are used to demonstrate the models' capability. The quality of calibration and validation results is not concerning at first sight, but should be placed in the context of other models' results - even though these are bound to be in other areas, for other timescales, etc. Only reporting such results leaves the reader with questions.

*So far, the SPHY-MMF model has not been applied in other catchments, therefore, we cannot report previous calibration and validation results on soil erosion and sediment yield. However, the hydrological model SPHY has been applied previously. Terink et al. (2015) reports several model applications in Romania, the Himalaya and Chile. We obtained similar calibration and validation results as reported by Terink et al. (2015). We have referred to these results in the manuscript. Comparison with performance of other erosion models would indeed be interesting, but only if applied to the same catchment and using the calibration strategy. Otherwise, we do not know the origin of the differences in performance, making objective interpretation of results of little value.*

During my review, I also missed reporting on results from the first (conservation agriculture) scenario – a climate change scenario appeared to get all the attention.

*The sustainable land management scenario was evaluated in section 3.3, where we applied reduced tillage to cereals and horticulture. The results are shown in Figure 7, indicated with the dashed lines. We have clarified this in the revised manuscript.*

Figures in the paper are good as it stands - although a bit more detail in them and their captions would make them more useful. I would appreciate another table with information on parameters (see my comment above). Detailed annotations and suggestions are available in the attached scan. All in all, I consider this an important contribution to literature presenting a model that usefully supplements the range of available models. I especially appreciate the authors' detailed and deliberate explanation of model equations. In light of the modest suggestions for improvement, I suggest minor revisions.

*We have added references to equations in the figure captions of Figures 1, 2 and 3.*

Page 2, line 28: By now you should become a bit more specific about which models you are thinking of. Perhaps a table works best.

*We prefer not to mention specific models at this stage, especially given the large number of available erosion and sediment models. Nevertheless, we have added reference to two model review papers to give an indication of what kind of models we are referring.*

Page 3, line 22: Introduction: well structured, clear, but insufficiently supported by references. The fact that there are many references to choose from, should not lead to you not showing any.

*We have now supported the introduction with specific references.*

Page 3, line 32: Specially at daily resolution, should the erosion model not feed back to SPHY as well? You erode and deposit top soil, with implications for infiltration and erodibility.

*Ideally a feedback from the soil erosion model to the hydrological model would result in changes to the top soil depth and subsequently its soil physical properties. However, given the fact that the model is intended to be applied at decadal time scales, the*

*erosion rates are not high enough (<1 mm yr$^{-1}$) to lead to substantial decreases of soil depth. Also, this could lead to problems when locally very high erosion rates are projected, eroding the top soil within the simulation period. For Landscape Evolution Models operating at longer time scales this would certainly be more relevant.*

Page 5, line 5: Does the single flow algorithm limit your model's usefulness to non-ploughed, i.e. non-smoothed, landscapes?

*We don't expect this to be the case, especially at the spatial resolution the model is intended to be applied (i.e. between 200 m and 1 km). A single flow might affect flow accumulation and concentrated flow processes in some cases, therefore we are actually working on a revised version with possibility of multiple flow simulation to prevent excessive concentration of flow.*

Page 5, line 7: I am very sympathetic of the brief model descriptions, so I like 2.2. However it gives me the idea that this 'physical-based model' still has many calibration parameters. Can you give us an overview of measurable / calibration parameters.

*We have included a table in the Supplementary Material, see previous comments.*

Page 8, line 25: Have these been introduced yet? How many are there?

*The soil texture classes have been introduced in the introduction paragraph of section 2.3. There are three soil texture classes, i.e. sand, silt and clay.*

Page 11, line 11: Does that not follow from Eq. 20-22 and the amount of water in the reservoir?

*Equations 20-22 determine how much sediment is deposited in the cell of its origin, i.e. immediate deposition of detached particles. The trapping efficiency of the reservoirs (Eq. 30) determines how much of the total amount of sediment that is transported from all the upstream cells is deposited in the reservoir, which is function of the capacity of the reservoir and the drainage area of the subcatchment.*

Page 12, line 1: With Ch. 2 in mind:
1. Great and calm explanation. I think I know what this model does
2. The model has many parameters. A table, perhaps in suppl. mat. Would help to distinguish measureable, literature based and calibration parameters.

*We have included a table in the Supplementary Material, see previous comments.*

Page 12, line 1: These figures are great but can be made more useful by referring to your equations in them.

*We have included references to the equation in the figure captions.*

Page 14, line 17: Why validate on the uncertain, old NDVI period, and calibrate on the good? Can you split more "fairly"?

*Indeed, it would be better to validate the model in a period where MODIS NDVI data are available. Unfortunately, these data are only available since 2000 and our climate forcing (precipitation and temperature) are only available until 2012. So we have a maximum of 13 'good' data years available. Especially for calibration and validation of the soil erosion model, a fairly long period is needed to include a number of large events which cause most erosion. When we would split the data into two equally sized periods, we would get two periods of 6 or 7 years, which would limit the occurrence of large events within these periods. Therefore, we choose to use a calibration period of 10 years and a validation period of 20 years, even though no NDVI was available for the validation period.*

Page 16, line 1: Table caption should clarify differences between cal/val target on one hand and Cerdan/Maetens number on other hand.

*We are uncertain about what the reviewer means with this comment. The table shows two columns with predicted hillslope erosion rates under calibration and under validation runs respectively, followed by two columns with published hillslope erosion rates by Cerdan/Maetens.*

Page 18, line 3:  You do not discuss the conservation agriculture scenario results. Or is that your 'reference' scenario? That would be confusing and should be improved.

*The sustainable land management (SLM) scenario was applied to two crops, i.e. cereals and horticulture, where we assumed that after harvest the fields are not tilled until the next sowing. We parameterized this by setting the plant height to 0 and leaving all other input parameters unchanged, which is different from the reference scenario. The results are shown in Figure 7, where the dashed lines show that soil erosion decreases under the SLM scenario. We have clarified this in section 3.3 and in the caption of Figure 7.*

Page 20, lines 2-7: This feels a little too much like an abstract or an advertisement. If in discussion, better at the end after you have provided the reader with the insight that supports it.

*We have removed this part from the manuscript.*

Page 20, lines 15-21: Well, you calibrated a few other parameters, and literature values are not always directly transferable. This aspect deserves more attention.

*We have added a discussion on the calibration procedure to the Discussion.*

Page 20, line 22: If that's your main finding, a much simpler model would do. Ample literature confirms this finding.

*This is certainly not our main finding, rather the main limitation of the current model, i.e. that the model projects unrealistic high erosion rates in the cells where most flow accumulates. We are currently working on an enhanced channel erosion module to fix this. We have changed this sentence accordingly.*

Page 21, line 2: Show/cite some examples from literature. People must have done this for a long time, I think.

*We have added citations to recent publications that have incorporated fluvial processes into soil erosion and landscape evolution models.*

**Referee #2**
First of all, I would like to congratulate the authors for the effort and dedication developed in this work. Any proposal of erosion modeling that goes beyond the estimation of potential annual erosive rates is an appreciable step forward for the scientific community, managers and, hence, the society. Reading the document has been easy, interesting and instructive to me.

*We would like to thank the reviewer for his nice comments on the manuscript.*

My general concerns are the following:
1) Studying the configuration of the model, with a clearly hillslope nature, I see its particular strength more related to the temporal scale of simulation, not so much with the spatial scale, and also with the role of vegetation on sediment transport and erosion processes. That is, both the considered dt and the adopted spatial resolution with its consequent limitations for the modeling of processes in Mediterranean environments can be justified by the model's ability to generate long time series based on processes and distributed information under different land use or climate scenarios. This is not reflected in the current manuscript and, in my opinion, its limitations are then not sufficiently justified.

*We agree that the temporal scale of the model is also a strength of the model, however, the model is intended to be applied at large spatial scales, from which the alternatives are mostly of empirical nature, such as WATEM-SEDEM, AGNPS, (R)USLE and SWAT (MUSLE). Most of these models do not include the most relevant hydrological and soil erosion processes that we explicitly consider in the our model, in particular intra-annual vegetation development and sediment deposition. Indeed, the considered daily time step and the adopted spatial resolution induce limitations for detailed soil erosion assessments. We have included a discussion on these issues in the Discussion section. Furthermore, the model has limitations in its capacity to accurately simulate concentrated flow, as also mentioned in the Discussion and in response to Reviewer 1. We are therefore working on an enhanced channel erosion module. See also the comments below regarding the limitations and considerations regarding extreme events and the spatial resolution of the model.*

2) The limitations of the model have to be declared and analyzed more in depth. Especially, I see problematic the modeling of extreme events, which are important in Mediterranean environments with frequent sub-hour time pulses. What is the implication of the runoff model

proposed in the results, as they indicate a considerable increase in this type of events in the considered future scenario?

*We agree, in some environments, like in the Mediterranean, sub-hourly rainfall amounts are often responsible for most soil erosion. However, often (sub-)hourly data are not available at the large spatial scale the model is intended to be applied. While (sub-)hourly data may be obtained from local meteorological stations, the data need to be interpolated to the model domain, which most likely results in large spatial and temporal uncertainties. Also, most climate model output is only available with a daily time step. A (sub-)hourly time step would limit the application of climate change scenarios and beyond events. Our results (Figures 8 and 9) show that the model is capable to simulate the increase of soil erosion by the expected increase in extreme precipitation. We have included a discussion on the limitations regarding the daily time step in the Discussion section.*

Has sub-hour rainfall data been analyzed? Could the model be modified to include these cases, ($\alpha$ = 1 at t = n and $\alpha$ = 0 at t $\neq$ n?), I see an attenuation effect by the model for this cases.

*The surface runoff equation (Eq. 8) includes a parameter that indicates the fraction of the daily rainfall that occurs in the hour with the highest intensity. This parameter could be set to 1, to indicate that all daily rainfall falls within one hour. However, rainfall in these environments is not only concentrated in one hour events and the case that the daily rainfall is more evenly spread over the day should also be considered. Although we used observed station data to determine the relation between daily and hourly precipitation, there is always a tradeoff between reality and model feasibility. The model is intended to be used at large spatial and temporal scales, which ultimately results in disregarding small scale phenomena (both spatial and temporal). For these phenomena a smaller scale model would be better suited.*

Also, I see inconsistencies, or I do not understand, the units in this part of the model ($\alpha$ in h$^{-1}$?, Qsurf mm/day?), please clarify.

*In the original manuscript we made some mistakes in the units for the variables used in Eq. 8. The infiltration rate f and precipitation P are in mm and $\alpha$ is a fraction and, therefore, dimensionless. This gives Qsurf in mm as well. We have changed the text accordingly.*

Another limitation is related to the selected cell size (200m grid size), Why this resolution? Is it related to the computational cost of the model? (I would like to know something about this issue), or is related to the remote sensing images? What limitations does this present from the point of view of the observed erosion processes in the study area, the forcing agents and their spatial distribution from the obtained results?

*The selected cell size is a tradeoff between the hydrological model and the soil erosion model. Indeed, for soil erosion processes a smaller cell size would be more appropriate. The selected cell size of 200 m is the lowest possible value the hydrological model allows, according to Terink et al. (2015). Some sub-soil processes, such as lateral flow and base flow, only act at large spatial scales. Therefore, the model has a lower limit*

*for the spatial resolution. We have included a discussion on this issue in the Discussion section.*

3) Fluvial vs hillslope contributions: the authors declare limitations of the model for modeling fluvial transport, which, in my opinion, should be assessed in the future from a fluvial sub-model that includes the basic erosion and transport processes (bedload + SS), integrating what that comes from the hillslopes (water and sediment) at different points. However, if I'm not wrong, the calibration/validation has been made from SSY estimated from reservoirs, what fraction corresponds to fluvial/hillslope contributions based on reservoir measurements?, this analysis is important and should be adressed given the process-based nature of the model.

*Indeed, the current model does treats erosion in channels the same as erosion on hillslopes . As stated above, we are working on a separate channel erosion module which would be part of a future model update (see Discussion section). However, in the current model, we do account for channel deposition processes, through the transport capacity equation (Eq. 28). Currently, this equation largely corrects for the high erosion rates generated by high accumulated flow rates in the river network. The current model is not able to distinguish between hillslope and fluvial contributions. In a future update of the model we would be able to perform such analysis.*

4) Although the document is well written and easy to read, the introduction seems too long and could be summarized.

*We think that the Introduction provides the essential background to better understand the processes included in the model and its intended use. Furthermore, Reviewer 1 commented that the Introduction is well structured and clear and asked for supporting references. Therefore, we have tried to streamline the introduction as much as possible without removing crucial information.*

On the contrary, more detailed information about the validation, calibration data, especially of the selected reservoirs (bathymetries, topographies, sediment grain sizes, ...) and SSY assessment is missing.

*Reservoir sediment yield was obtained from literature (Avendaño-Salas et al., 1997), which included bathymetric data on changes in reservoir volume and bulk density of deposited sediment, from which we determined the yearly sediment yield. Hillslope erosion (SSY) was also calibrated using literature values (Cerdan et al., 2010; Maetens et al., 2012). We have clarified the calibration procedure in section 3.2.*

It may be interesting to incorporate a section of uncertainty analysis.

*As suggested by reviewer 1, instead of a full sensitivity analysis, which would make the paper too long at this stage, we have added a table to the Supplementary Material that includes all model parameters mentioned in the manuscript. The table indicates the following aspects: landuse-specific, measurable, calibration, literature based and a citation where we obtained the literature values of the parameters.*

Please, check the citation format of the entire document.

*We have changed the citation format.*

In general, I encourage the authors to highlight the most interesting aspects of the model, taking into account the spatial and temporal scales adopted, as well as reasonably stating of the associated limitations in order to evaluate not only the model, but all the related challenges. Some minor comments are reported in the attached file.

Page 4, line 11: Please, specify satellite platform and data correction used.

*The user could provide any NDVI images, as long as they are in the same resolution and within the domain as the model. In the model application we used MODIS NDVI images, but the model is not limited to this specific dataset. Therefore, we do not specify here which satellite platform should be used, this is up to the user.*

Page 9, line 1: Put this values also at the context studied by Knapen et al. (Resistance of soils to concentrated flow erosion: A review).

*These values refer to the detachability of the soil by raindrop impact (K), rather than detachability of the soil by runoff (DR), which is used in Eq. 19. Both K and DR cannot directly be obtained from Knapen et al. (2007), which consider the concentrated flow erodibility and critical flow shear stress. Equations 18 and 19 differ from the soil detachment equations discussed in Knapen et al. (2007), which include several process-based approaches, such as excess shear stress, excess stream power and transport capacity deficit approaches. These approaches either use concentrated flow erodibility or critical flow shear stress to quantify the detachability of the soil by runoff. The SPHY-MMF model uses a different approach, which may be classified as an intermediate between empirical and process-based approaches. Therefore, the values reported in Knapen et al. (2007) are not directly comparable with the values for K and DR.*

Page 18, line 11: Are you sure that the model is capturing the increase of heavy precipitation? How this can condition these results? Please stress the limitation of the model and uncertainties.

*Figure 8 shows that an increase of heavy precipitation is expected under the future climate change scenario. This results in an increase of soil erosion as well (Figure 9). However, this does not lead to an increase of reservoir sediment yield, but a decrease. We argue that this is caused by a decrease of runoff, which causes a decrease of transport capacity (see Eq. 28).*

[revised manuscript text omitted]

**Table S1.** Input parameters for the model. Parameter determination indicates if the parameter can be obtained from literature (L), is measurable (M) or should be obtained from calibration (C). The bold characters indicate how the parameter values were obtained in the model application.

| description | symbol | unit | equation | landuse-specific | parameter determination | reference |
|---|---|---|---|---|---|---|
| **hydrological model** | | | | | | |
| depletion fraction | $p_{\text{tabular}}$ | - | 4 | × | **L** | Allen et al. (1998, Table 22) |
| crop coefficient open-water evaporation | $kc_{\text{open-water}}$ | - | 5 | | **L** / C | Allen et al. (1998) |
| calibration parameter infiltration rate | $\lambda$ | - | 6 | | **C** | |
| fraction of daily rainfall | $\alpha$ | - | 8 | | **M / C** | |
| maximum LAI | $LAI_{\text{max}}$ | - | 9 | × | **L** | Sellers et al. (1996) |
| **soil erosion model** | | | | | | |
| plant height | $PH$ | m | 13 | × | **L / M / C** | |
| intensity of erosive precipitation | $I$ | mm h$^{-1}$ | 14 | | **L** / C | Morgan and Duzant (2008) |
| canopy cover[1] | $CC$ | - | 16 | × | **M** / C | |
| detachability by raindrop impact | $K$ | g J$^{-1}$ | 18 | | **L** / M / C | Quansah (1982) |
| detachability by runoff | $DR$ | g mm$^{-1}$ | 19 | | **L** / M / C | Quansah (1982) |
| ground cover | $GC$ | - | 18, 19 | × | **M / C** | |
| water depth | $d$ | m | 21, 23, 26 | | **L** / C | Morgan and Duzant (2008) |
| sediment density | $\rho_s$ | kg m$^{-3}$ | 22 | | **L / M** | Morgan and Duzant (2008) |
| flow density | $\rho$ | kg m$^{-3}$ | 22 | | **L / M** | Morgan and Duzant (2008) |
| fluid veiscosity | $\eta$ | kg m$^{-1}$ s$^{-1}$ | 22 | | **L / M** | Morgan and Duzant (2008) |
| diameter of soil particles | $\delta$ | m | 22 | | **L / M** | Morgan and Duzant (2008) |
| Manning's roughness bare soil | $n_{\text{soil}}$ | s m$^{-1/3}$ | 24 | | **L** / C | Morgan and Duzant (2008) |
| Manning's roughness vegetation | $n_{\text{vegetation}}$ | s m$^{-1/3}$ | 24 | × | **L** / C | Chow (1959) |
| surface roughness for tilled soil | $RFR$ | cm m$^{-1}$ | 25 | | **L** / C | Morgan and Duzant (2008) |
| stem diameter | $D$ | m | 26 | × | **L** / M / **C** | |
| stem density | $NV$ | stems m$^{-2}$ | 26 | × | **M / C** | |
| **sediment transport** | | | | | | |
| parameter transport capacity 1 | $\beta$ | - | 28 | | L / **C** | Prosser and Rustomji (2000) |
| parameter transport capacity 2 | $\gamma$ | - | 28 | | L / **C** | Prosser and Rustomji (2000) |
| water depth bare soil | $d_{\text{bare}}$ | m | 29 | | **L** / C | Morgan and Duzant (2008) |
| water depth transport capacity | $d_{\text{actual}}$ | m | 29 | | **L** / C | Morgan and Duzant (2008) |
| trapping efficiency constant | $D$ | - | 30 | | **L** / C | Brown (1943) |

[1] can be obtained from NDVI